# Self-Correcting Decoding with Generative Feedback for Mitigating Hallucinations in Large Vision-Language Models

**Ce Zhang**[*1]  **Zifu Wan**[*1]  **Zhehan Kan**[2]  **Martin Q. Ma**[1]  **Simon Stepputtis**[1]
**Deva Ramanan**[1]  **Russ Salakhutdinov**[1]  **Louis-Philippe Morency**[1]  **Katia Sycara**[1]  **Yaqi Xie**[1]
[1]School of Computer Science, Carnegie Mellon University
[2]Shenzhen International Graduate School, Tsinghua University

## Abstract

While recent Large Vision-Language Models (LVLMs) have shown remarkable performance in multi-modal tasks, they are prone to generating hallucinatory text responses that do not align with the given visual input, which restricts their practical applicability in real-world scenarios. In this work, inspired by the observation that the text-to-image generation process is the inverse of image-conditioned response generation in LVLMs, we explore the potential of leveraging text-to-image generative models to assist in mitigating hallucinations in LVLMs. We discover that generative models can offer valuable self-feedback for mitigating hallucinations at both the response and token levels. Building on this insight, we introduce self-correcting Decoding with Generative Feedback (DeGF), a novel training-free algorithm that incorporates feedback from text-to-image generative models into the decoding process to effectively mitigate hallucinations in LVLMs. Specifically, DeGF generates an image from the initial response produced by LVLMs, which acts as an auxiliary visual reference and provides self-feedback to verify and correct the initial response through complementary or contrastive decoding. Extensive experimental results validate the effectiveness of our approach in mitigating diverse types of hallucinations, consistently surpassing state-of-the-art methods across six benchmarks. Code is available at https://github.com/zhangce01/DeGF.

## 1 Introduction

Large Vision-Language Models (LVLMs) have demonstrated remarkable performance across various multi-modal tasks, such as image captioning and visual question answering, by extending the capabilities of powerful Large Language Models (LLMs) to incorporate visual inputs (Liu et al., 2023; Li et al., 2023b; Dai et al., 2023; Bai et al., 2023; Ye et al., 2024). Despite their proficiency in interpreting both visual and textual modalities, these models often suffer from *hallucinations*, where LVLMs erroneously produce responses that are inconsistent with the visual input (Li et al., 2023d; Gunjal et al., 2024; Yin et al., 2023; Wu et al., 2024). This potential for misinformation raises significant concerns, limiting the models' reliability and restricting their broader deployment in real-world scenarios (Liu et al., 2024b; Bai et al., 2024; Chen et al., 2024b; Zhao et al., 2024).

Recent research has revealed that a major cause of hallucinations in LVLMs is the over-reliance on language priors due to biased training sets, which can override the visual content in response generation (Bai et al., 2024; Liu et al., 2024b; Leng et al., 2024). In response, various strategies have been developed to detect and mitigate these hallucinations by directly introducing additional training (Chen et al., 2024a; Sun et al., 2023; Jiang et al., 2024; Chen et al., 2023; Zhang et al., 2024), demonstrating promising results in reducing over-reliance. However, the need for additional data and costly training processes hinders their deployment in downstream tasks. More recently, a new paradigm of methods has emerged to tackle the hallucination problem in LVLMs by intervening in the decoding process (Huang et al., 2024; Deng et al., 2024; Kim et al., 2024). Among these, recent training-free contrastive decoding-based methods (Li et al., 2023c) have proven effective in mitigating undesired hallucinations by contrasting token predictions derived from original visual input with bias-inducing counterparts, such as no/distorted visual input (Favero et al., 2024; Leng et al., 2024), disturbed instructions (Wang et al., 2024), or premature layers (Chuang et al., 2024).

---

*Equal contribution. †Contact: {cezhang, zifuw, yaqix}@cs.cmu.edu.

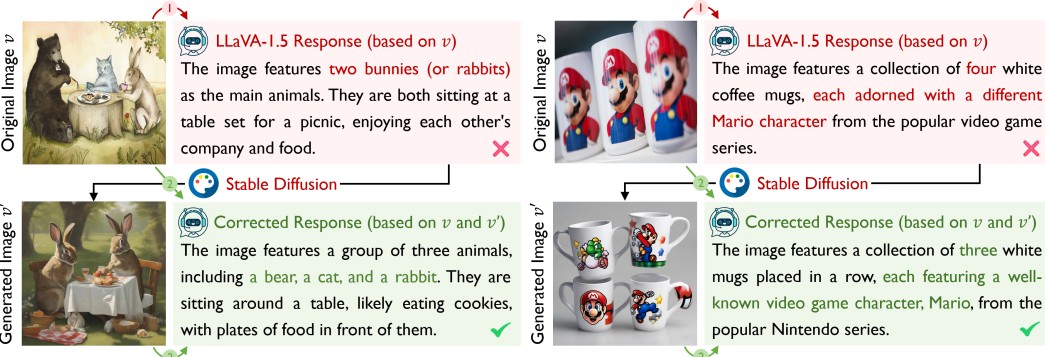

Figure 1: **Generative models can visualize and help correct various types of hallucinations in the initial response**. ① In the first query, we provide LLaVA-1.5 (Liu et al., 2023) with the prompt "`Describe this image in detail`" to produce captions for two examples from LLaVA-Bench. Based on the initial response, we utilize Stable Diffusion XL (Podell et al., 2024) to generate a new image $v'$, which effectively highlights hallucinations and provides valuable self-feedback. ② In the second query, our approach incorporates both the original image $v$ and the generated image $v'$ into the decoding process, using the feedback to successfully correct various types of hallucinations.

While these contrastive decoding-based methods effectively mitigate hallucinations arising from language priors, we recognize that hallucinations can also originate beyond language bias, stemming from visual deficiencies in LVLMs (Tong et al., 2024). For instance, in counting hallucinations, language does not imply any count information; instead, miscounts largely arise from visual recognition errors of LVLMs, as complex scenes include numerous, similar objects at ambiguous positions which may confuse the LVLMs, leading to incorrect visual understanding and, consequently, hallucinated answers. Therefore, we argue that current contrastive decoding-based methods may struggle to generalize effectively across different types of hallucinations.

In this work, we explore the potential of leveraging powerful text-to-image generative models (*e.g.*, Stable Diffusion (Rombach et al., 2022; Podell et al., 2024)) to mitigate various types of hallucinations in LVLMs. Our work is based on a simple yet intuitive hypothesis: Given a visual input and a textual prompt to an LVLM, if the generated response conditioned on the original image is accurate and non-hallucinatory, a text-to-image generative model should be capable of reversing this process to produce a similar image from that response. Alternatively, if there is a discrepancy between the original image and the one generated from the response, this difference can serve as valuable self-feedback, guiding the decoding process to correct potential hallucinations in the initial response. To verify this hypothesis, we conduct an empirical study (in Section 3.2), demonstrating that *generative models can provide valuable self-feedback for mitigating hallucinations at both the response and token levels*.

Building on this insight, we introduce self-correcting Decoding with Generative Feedback (DeGF), a novel training-free decoding algorithm that effectively incorporates feedback from text-to-image generative models to recursively enhance the accuracy of LVLM responses. Specifically, for each instance, we generate a new image based on the initial response, which serves as an *auxiliary visual reference* to assess and verify the accuracy of the initial output. We propose self-correcting decoding that either enhances or contrasts predictions from the original and this reference based on the auxiliary visual reference, *confirming* or *revising* the initial LVLM response based on the degree of divergence between the two predictions. By integrating this additional visual reference and generative feedback, LVLMs can gain enhanced visual insights and verify the initial response to ensure accurate visual details in the text outputs. In Figure 1, we demonstrate that incorporating generative feedback in our approach can reduce various types of hallucinations, including object existence, visual appearance, counting, *etc*. To the best of our knowledge, we are the first work to explore the use of text-to-image generative feedback as a self-correcting mechanism for mitigating hallucinations in LVLMs.

The effectiveness of DeGF is evaluated on LLaVA-1.5, InstructBLIP, and Qwen-VL across six benchmarks: POPE (Li et al., 2023d), CHAIR (Rohrbach et al., 2018), MME-Hallucination (Fu et al., 2023), MMBench (Liu et al., 2024d), MMVP (Tong et al., 2024), and LLaVA-Bench. Extensive experimental results validate the effectiveness of our DeGF in mitigating various types of hallucinations in LVLMs. Qualitative case studies and GPT-4V-aided evaluation on LLaVA-Bench further demonstrate that our approach enhances both the accuracy and detailedness of the LVLM responses.

The contributions of this paper are summarized as follows:

- We investigate the potential of text-to-image generative models in mitigating hallucinations in LVLMs and demonstrate that text-to-image generative models can provide valuable self-feedback for mitigating hallucinations at both the response and token levels.
- We propose self-correcting Decoding with Generative Feedback (DeGF), a novel training-free decoding algorithm for LVLMs that recursively enhances the accuracy of responses by integrating feedback from text-to-image generative models with complementary/contrastive decoding.
- Extensive experimental evaluations across six benchmarks demonstrate that our DeGF consistently outperforms state-of-the-art approaches in effectively mitigating hallucinations in LVLMs.

## 2 RELATED WORK

**Hallucination in LVLMs**. With advances of autoregressive LLMs (Touvron et al., 2023; Chowdhery et al., 2023; Chiang et al., 2023), researchers have extended these powerful models to process visual inputs, leading to the development of LVLMs (Liu et al., 2023; Dai et al., 2023; Bai et al., 2023; Ye et al., 2024). These models typically train a modality alignment module to project visual tokens into the textual embedding space of the LLM, demonstrating impressive performance in various multi-modal tasks such as visual question answering and image captioning (Liu et al., 2024b; Bai et al., 2024). However, LVLMs are prone to hallucinations, where contradictions arise between the visual content and the generated textual response (Li et al., 2023d; Liu et al., 2024b; Bai et al., 2024).

To mitigate hallucinations in LVLMs, early works have introduced various approaches, including reinforcement learning from human feedback (RLHF) (Gunjal et al., 2024; Sun et al., 2023), applying auxiliary supervision (Jiang et al., 2024; Chen et al., 2023), incorporating negative (Liu et al., 2024a) or noisy data (Yue et al., 2024), and training post-hoc revisors for correction (Zhou et al., 2024; Yin et al., 2023). Despite promising results, these methods often lack practicality due to their reliance on additional data and costly training processes. To address this, another line of work focuses on training-free methods that can be seamlessly integrated into existing LVLMs. Such methods encompass contrastive decoding (Leng et al., 2024; Favero et al., 2024) and guided decoding with auxiliary information (Chen et al., 2024d; Deng et al., 2024; Woo et al., 2024). In this work, we present a novel training-free approach that recursively enhances the accuracy of the LVLM response by incorporating text-to-image generative feedback. To the best of our knowledge, we are the first work to effectively utilize feedback from text-to-image generative models to mitigate hallucinations in LVLMs.

**Text-to-Image Synthesis**. Text-to-image synthesis aims to create realistic images from textual descriptions (Zhu et al., 2019; Ge et al., 2023). In recent years, significant progress has been achieved in this area, largely due to the advent of deep generative models (Zhan et al., 2023; Goodfellow et al., 2014). These advances include Generative Adversarial Networks (GAN) (Sauer et al., 2023; Kang et al., 2023), autoregressive models (Chang et al., 2023; Yu et al., 2022), and diffusion models (Ho et al., 2020; Karras et al., 2022; Nichol et al., 2022; Saharia et al., 2022; Rombach et al., 2022). Among these, diffusion-based methods have been particularly distinguished due to their ability to generate high-quality, detailed images with fine-grained control over the synthesis process (Yang et al., 2023; Croitoru et al., 2023). Pre-trained on large-scale text-image datasets such as LAION (Schuhmann et al., 2022), diffusion-based methods have demonstrated strong vision-language alignment, making them valuable for downstream tasks such as classification (Li et al., 2023a) and semantic segmentation (Amit et al., 2021; Wolleb et al., 2022).

More recently, Jiao et al. (2024) incorporate text-to-image generative models to enhance fine-grained image recognition in LVLMs by introducing the Img-Diff dataset, which generates pairs of similar images using Stable Diffusion XL (Podell et al., 2024). Their results demonstrate that fine-tuning LVLMs with this additional data leads to improved performance on several VQA tasks. In contrast, in this work, we directly leverage a pre-trained diffusion model to provide valuable self-feedback for refining the generated responses of LVLMs in the decoding process, dynamically improving the accuracy and consistency of the model's response without modifying the underlying LVLMs.

## 3 METHOD

In this work, we present DeGF, a novel training-free algorithm that recursively improves the accuracy of LVLM responses using text-to-image generative feedback, as illustrated in Figure 2.

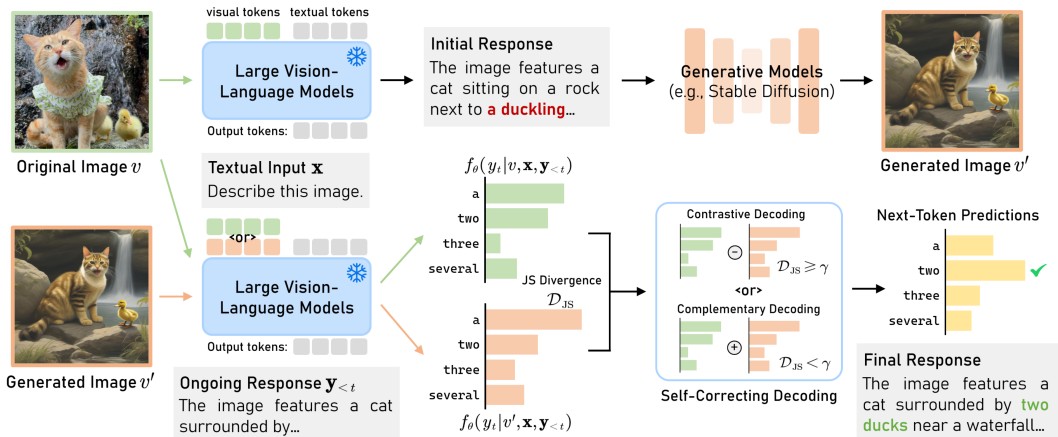

Figure 2: **Overview of our proposed DeGF**. Our method follows a two-step process: first, a generative model produces a high-quality image based on the initial response; second, this image acts as an auxiliary visual reference, providing feedback to refine the next-token predictions. Additionally, we introduce self-correcting decoding, which either enhances or contrasts the next-token predictions conditioned on the original and generated images to mitigate hallucinations in the LVLM response.

## 3.1 PRELIMINARY: DECODING OF LVLMS

We consider an LVLM parameterized by $\theta$, which processes an input image $v$ and a textual query $\mathbf{x}$, aiming to autoregressively generate a fluent sequence of textual responses $\mathbf{y}$. The visual input $v$ is first processed by a vision encoder and then projected into visual tokens within the textual input space using a vision-language alignment module (*e.g.*, Q-Former (Li et al., 2023b) or linear projection (Liu et al., 2023)). These visual tokens, along with the textual query tokens, are then fed into the language encoder for conditioned autoregressive generation. We denote the autoregressive generation process as

$$y_t \sim p_\theta(y_t|v, \mathbf{x}, \mathbf{y}_{<t}) \propto \exp f_\theta(y_t|v, \mathbf{x}, \mathbf{y}_{<t}), \tag{1}$$

where $y_t$ represents the token at time step $t$, $\mathbf{y}_{<t} \triangleq [y_0, \ldots, y_{t-1}]$ denotes the sequence of tokens generated before time step $t$, and $f_\theta$ is the logit distribution (unnormalized log-probabilities) produced by the LVLM over a vocabulary of textual tokens $\mathcal{V}$. At each step $t \in [0, \ldots, T]$, the response token $y_t$ is sampled from the probability distribution $p_\theta(y_t|v, \mathbf{x}, \mathbf{y}_{<t})$, and this generative process continues iteratively until the response sequence $\mathbf{y} \triangleq [y_0, \ldots, y_T]$ is complete.

## 3.2 VISUAL REFERENCE GENERATION

In our method, we incorporate generative feedback from diffusion models to guide the decoding process. Specifically, given a visual input $v$ and a textual query $\mathbf{x}$, we first prompt the LVLMs to generate an initial response $\boldsymbol{\tau}$, which includes relevant descriptions of the visual input with potential hallucinations. Subsequently, we leverage a pre-trained diffusion model $\mathcal{G}$ to generate a new image $v'$ based on the initial response:

$$v' = \mathcal{G}(\boldsymbol{\tau}, x_T), \quad \text{where } x_T \sim \mathcal{N}(0, \mathbf{I}). \tag{2}$$

Here, $x_T$ denotes a sample from the standard Gaussian distribution, which serves as the initial noisy input to the diffusion model. Starting from this pure noise image $x_T$, the diffusion model $\mathcal{G}$ iteratively applies $T$ steps of the denoising process to obtain $x_T, x_{T-1}, \ldots, x_0$, where the final output $x_0$ corresponds to the final generated image $v'$. Through this diffusion process, the generative model visualizes the initial response, providing a visual reference that helps mitigate potential hallucinations and produce a more accurate and consistent output.

**Effectiveness of Text-to-Image Generative Models in Reflecting Hallucinations**. We validate the effectiveness of generative models in reflecting hallucinations through an empirical study, as shown in Figure 3.[1] The experimental results demonstrate that *text-to-image generative models can provide valuable self-feedback for mitigating hallucinations* at both the response and token levels.

---

[1]For Figure 3, we evaluate 1,000 CHAIR samples (*Left*) and 3,000 POPE samples (*Right*).

We conduct the following two experiments: (1) We generate an image $v'$ using diffusion model based on the initial caption provided by LLaVA-1.5 and compute the CLIP image similarities between the original image $v$ and the generated image $v'$ using OpenCLIP (Cherti et al., 2023) ViT-H/14 backbone. Following prior work, we use the CHAIR (Rohrbach et al., 2018) benchmark, a rule-based metric on MS-COCO (Lin et al., 2014) for evaluating object hallucination from generated captions. We report the average per-instance metric $CHAIR_I$ within each bin of CLIP similarity, which evaluates the object hallucination rates in the entire initial response.

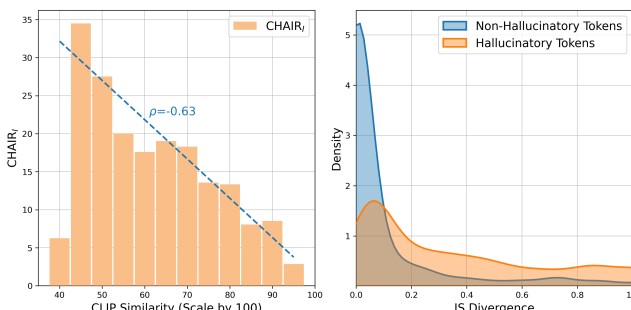

Figure 3: **Text-to-image generative models can provide feedback for reflecting hallucinations**. (*Left*) Bar plot of average $CHAIR_I$ scores binned by CLIP similarity (scaled by 100) on the CHAIR benchmark; (*Right*) Density plots of token-level JS divergence for both hallucinatory and non-hallucinatory tokens on the POPE benchmark.

As shown in Figure 3 (*Left*), a clear negative correlation between hallucination rates and CLIP similarities is observed (with a correlation coefficient of $\rho = -0.63$). This indicates that *lower similarity between the original image and generated image corresponds to higher rates of hallucinations at the response level*. (2) Similarly, we generate an image $v'$ based on the initial response given by LLaVA-1.5 for each instance on the POPE (Li et al., 2023d) benchmark. In Figure 3 (*Right*), we present the density plot of Jensen-Shannon (JS) divergence between the predicted probabilities for both images, *i.e.*, $p_\theta(y_t|v, \mathbf{x}, \mathbf{y}_{<t})$ and $p_\theta(y_t|v', \mathbf{x}, \mathbf{y}_{<t})$, for hallucinatory and non-hallucinatory tokens.[2] The results show that the density of JS divergence follows a long-tail distribution, with hallucinatory tokens exhibiting significantly longer tails and higher JS divergence. This shows *JS divergence between probabilities derived from the original and the generated image corresponds well to hallucinations at the token level.* These observations provide insights into the effectiveness of generative models in reflecting hallucinations, and motivate us to incorporate the generative feedback during the decoding process.

## 3.3 SELF-CORRECTING DECODING WITH GENERATIVE FEEDBACK

In this section, we focus on effectively utilizing generative feedback during the decoding process to mitigate potential hallucinations. Specifically, we propose a self-correcting decoding approach that leverages generative feedback to *confirm* or *revise* the initial response by selectively enhancing or contrasting the logits for each generated token based on the measured divergence between the two predicted probability distributions.

Specifically, to predict a specific token $y_t$, we utilize LVLMs to generate two output distributions, each conditioned on either the original image $v$ or the synthesized visual reference $v'$, expressed as:

$$p_\theta(y_t|v, \mathbf{x}, \mathbf{y}_{<t}) = \texttt{Softmax}[f_\theta(y_t|v, \mathbf{x}, \mathbf{y}_{<t})], \quad p_\theta(y_t|v', \mathbf{x}, \mathbf{y}_{<t}) = \texttt{Softmax}[f_\theta(y_t|v', \mathbf{x}, \mathbf{y}_{<t})].$$
(3)

We define and compute the following distance metric based on Jensen-Shannon (JS) divergence at each timestep $t$ to quantify the discrepancy between two next-token probability distributions:

$$d_t(v, v') = \mathcal{D}_{\text{JS}}\left(p_\theta\left(y_t|v, \mathbf{x}, \mathbf{y}_{<t}\right) \| p_\theta\left(y_t|v', \mathbf{x}, \mathbf{y}_{<t}\right)\right),$$

$$\text{where } \mathcal{D}_{\text{JS}}(P \| Q) = \frac{1}{2}\mathcal{D}_{\text{KL}}(P \| M) + \frac{1}{2}\mathcal{D}_{\text{KL}}(Q \| M), \text{ and } M = \frac{1}{2}(P + Q).$$
(4)

Here, $\mathcal{D}_{\text{KL}}$ represents the Kullback-Leibler (KL) divergence. Note that $d_t(v, v') \in [0, 1]$ is a symmetric metric, providing a fine-grained measure of how closely the two distributions align as the model predicts each subsequent token.

We consider two scenarios based on the token-level generative feedback: (1) If the two predictions are aligned and both images agree on a specific token prediction, we *confirm* the original prediction as correct, and the auxiliary prediction from the generated image can be combined with the original

---

[2]Note that POPE benchmark contains yes-or-no questions about object existence. In this experiment, we evaluate only the first response token (*i.e.*, yes or no) to determine the presence of hallucinations.

prediction for enhancement (complementary decoding (Woo et al., 2024)). (2) Conversely, if there is a significant discrepancy between the predictions, indicating that the original prediction is likely hallucinatory, we *revise* the original response by using the generated visual input as a contrasting reference to refine the initial next-token prediction (contrastive decoding (Leng et al., 2024)). To implement this, we introduce a distance threshold $\gamma$ and develop two corresponding decoding approaches as follows:

$$
y_t \sim p_\theta(y_t) = \begin{cases} \texttt{Softmax}\left[f_\theta(y_t|v, \mathbf{x}, \mathbf{y}_{<t}) + \alpha_1\, f_\theta(y_t|v', \mathbf{x}, \mathbf{y}_{<t})\right], & \text{if } d_t(v, v') < \gamma; \\ \texttt{Softmax}\left[(1 + \alpha_2)\, f_\theta(y_t|v, \mathbf{x}, \mathbf{y}_{<t}) - \alpha_2\, f_\theta(y_t|v', \mathbf{x}, \mathbf{y}_{<t})\right], & \text{if } d_t(v, v') \geq \gamma, \end{cases}
$$
(5)

where $\alpha_1$ and $\alpha_2$ are hyperparameters that control the influence of the generated visual reference in the final prediction. Note that setting $\alpha_1 = 0$ or $\alpha_2 = 0$ degrades this process to regular decoding. The final generated token $y_t$ is sampled from the multinomial distribution with probabilities $p_\theta(y_t)$.

## 4 EXPERIMENTS

In this section, we evaluate the effectiveness of our method in mitigating hallucinations in LVLMs across a range of benchmarking scenarios, comparing it with existing state-of-the-art approaches.

### 4.1 EXPERIMENTAL SETTINGS

**Evaluated LVLMs**. We evaluate the effectiveness of our method on three state-of-the-art open-source LVLMs: LLaVA-1.5 (Liu et al., 2024c), InstructBLIP (Dai et al., 2023), and Qwen-VL (Bai et al., 2023). Both LLaVA-1.5 and InstructBLIP utilize Vicuna-7B (Chiang et al., 2023) as the language encoder, which is instruction-tuned from LLaMA (Touvron et al., 2023). In contrast, Qwen-VL (Bai et al., 2023) is based on the Qwen 7B backbone. Specifically, we implement our approach using weights of the Qwen-VL-Chat model.

**Benchmarks**. We conduct extensive experiments on six benchmarks: (1) **POPE (Li et al., 2023d)** is a widely used benchmark for assessing object hallucinations in LVLMs, which tests the models with yes-or-no questions regarding the presence of specific objects, such as, "Is there a {object} in the image?" (2) **CHAIR (Rohrbach et al., 2018)** evaluates object hallucinations in open-ended captioning tasks. It prompts the LVLMs to describe specific images selected from a random sample of 500 images from the MSCOCO validation set; (3) **MME-Hallucination (Fu et al., 2023)** is a comprehensive benchmark for LVLMs consisting of four subsets: *existence* and *count* for object-level hallucinations, and *position* and *color* for attribute-level hallucinations; (4) **MMBench (Liu et al., 2024d)** serves as a comprehensive benchmark designed to assess the multi-modal understanding capabilities of LVLMs across 20 dimensions; (5) **MMVP (Tong et al., 2024)** collects CLIP-blind pairs and evaluates the fine-grained visual recognition capabilities of LVLMs. It consists of 150 image pairs, each accompanied by a binary-option question; (6) **LLaVA-Bench** provides 24 images featuring complex scenes, memes, paintings, and sketches, along with 60 challenging questions.

**Baselines**. As a simple baseline, we include results from regular decoding, where the next token is sampled directly from the post-softmax probability distribution. Additionally, we compare the performance of our method with three state-of-the-art decoding approaches: VCD (Leng et al., 2024), M3ID (Favero et al., 2024), and RITUAL (Woo et al., 2024). For evaluations on the CHAIR (Rohrbach et al., 2018) and MME-Hallucination (Fu et al., 2023) benchmark, we further include comparisons with Woodpecker (Yin et al., 2023), HALC (Chen et al., 2024d), DoLa (Chuang et al., 2024) and OPERA (Huang et al., 2024). We report the performance of these baselines based on our re-implementation using their released code bases.

**Implementation Details**. In our experiments, we adhere to the default query format for the input data used in both LLaVA-1.5 (Liu et al., 2024c) and InstructBLIP (Dai et al., 2023). Additionally, we set $\alpha_1 = 3$, $\alpha_2 = 1$, and $\gamma = 0.1$ by default in our decoding process. We follow VCD (Leng et al., 2024) to implement adaptive plausibility constraints (Li et al., 2023c), where we set $\beta = 0.1$ in open-ended CHAIR benchmark and $\beta = 0.25$ for other tasks. To ensure the reliability of our results, we conduct MME experiments three times with different initialization seeds and report the mean accuracy along with the standard deviation. All experiments are conducted on a single 48GB NVIDIA RTX 6000 Ada GPU. More implementation details are provided in Section B of the Appendix.

Table 1: **Results on POPE (Li et al., 2023d) benchmark**. Higher (↑) accuracy, precision, recall, and F1 indicate better performance. The best results are **bolded**, and the second-best are underlined.

| | Setup | Method | LLaVA-1.5 | | | InstructBLIP | | | Qwen-VL | | |
|---|---|---|---|---|---|---|---|---|---|---|---|
| | | | Acc. ↑ | Prec. ↑ | F1 ↑ | Acc. ↑ | Prec. ↑ | F1 ↑ | Acc. ↑ | Prec. ↑ | F1 ↑ |
| MS-COCO | Random | Regular | 83.13 | 81.94 | 83.44 | 83.07 | 83.02 | 83.08 | 87.43 | 93.56 | 86.48 |
| | | VCD | 87.00 | 86.13 | 87.15 | 86.23 | 88.14 | 85.88 | 88.80 | 93.89 | 88.11 |
| | | M3ID | 87.50 | 87.38 | 87.52 | 86.67 | 88.09 | 86.41 | **89.83** | 95.44 | 89.17 |
| | | RITUAL | 88.87 | 89.23 | **88.81** | **88.83** | 90.48 | **88.60** | 89.47 | **96.32** | 88.62 |
| | | **Ours** | **89.03** | **91.20** | 88.74 | **88.83** | 93.73 | 87.71 | 89.73 | 93.19 | **89.31** |
| | Popular | Regular | 81.17 | 78.28 | 82.08 | 77.00 | 73.82 | 78.44 | 84.70 | 88.24 | 83.96 |
| | | VCD | 83.10 | 79.96 | 83.94 | 80.07 | 77.67 | 80.89 | 85.13 | 87.27 | 84.69 |
| | | M3ID | 84.30 | 81.58 | 84.95 | 80.97 | 77.93 | 81.85 | 86.27 | 89.19 | **85.73** |
| | | RITUAL | 85.83 | 84.17 | 86.17 | 81.97 | 78.90 | **82.87** | 84.57 | 84.09 | 84.67 |
| | | **Ours** | **86.63** | **87.75** | **86.28** | **82.73** | **84.02** | 82.10 | **86.50** | **89.87** | 85.71 |
| | Adversarial | Regular | 77.43 | 73.31 | 79.26 | 74.60 | 71.26 | 76.45 | 79.83 | 80.13 | 79.73 |
| | | VCD | 77.17 | 72.18 | 79.47 | 77.20 | 74.29 | 78.49 | 81.33 | 80.60 | 81.55 |
| | | M3ID | 78.23 | 73.51 | 80.22 | 77.47 | 73.68 | 79.14 | 82.03 | 81.47 | 82.19 |
| | | RITUAL | 78.80 | 74.43 | 80.54 | 78.73 | 74.57 | **80.39** | 82.80 | 83.15 | 82.71 |
| | | **Ours** | **81.63** | **80.59** | **81.94** | **80.30** | **80.90** | 80.11 | **83.47** | **84.49** | **82.98** |
| A-OKVQA | Random | Regular | 81.90 | 76.63 | 83.53 | 80.63 | 76.82 | 81.92 | 86.27 | 90.66 | 85.48 |
| | | VCD | 83.83 | 78.05 | 85.34 | 84.20 | 80.90 | 85.00 | 87.87 | 90.06 | 87.53 |
| | | M3ID | 84.67 | 79.25 | 85.97 | 85.43 | 81.77 | 86.23 | **88.13** | 92.06 | 87.55 |
| | | RITUAL | 85.17 | 79.79 | 86.40 | 87.13 | 83.92 | **87.71** | 87.73 | **92.49** | 87.01 |
| | | **Ours** | **86.93** | **84.28** | **87.42** | **87.40** | **87.67** | 87.26 | 87.90 | 89.16 | **87.58** |
| | Popular | Regular | 75.07 | 68.58 | 78.77 | 75.17 | 70.15 | 77.91 | 84.60 | 87.99 | 83.88 |
| | | VCD | 76.63 | 69.59 | 80.19 | 78.63 | 73.53 | 80.72 | 86.23 | 87.30 | 86.03 |
| | | M3ID | 77.80 | 70.98 | 80.91 | 78.80 | 73.38 | 81.00 | **86.50** | 89.59 | 85.95 |
| | | RITUAL | 78.83 | 71.99 | 81.68 | 78.73 | 72.83 | 81.17 | 86.36 | 88.73 | 86.20 |
| | | **Ours** | **80.90** | **75.68** | **82.66** | **81.47** | **78.61** | **82.35** | 86.43 | **90.74** | **86.52** |
| | Adversarial | Regular | 67.23 | 61.56 | 73.70 | 69.87 | 64.54 | 74.54 | 76.90 | 75.59 | 77.48 |
| | | VCD | 67.40 | 61.39 | 74.21 | 71.00 | 65.41 | 75.45 | 79.13 | 76.04 | 80.30 |
| | | M3ID | 68.60 | 62.22 | 75.11 | 70.10 | 64.28 | 75.16 | 79.50 | 77.54 | 80.21 |
| | | RITUAL | 68.57 | 62.26 | 74.99 | 70.27 | 64.15 | 75.55 | 80.20 | 79.08 | **80.58** |
| | | **Ours** | **72.70** | **66.70** | **76.86** | **73.93** | **69.36** | **76.67** | **80.75** | **80.37** | 80.46 |
| GQA | Random | Regular | 82.23 | 76.32 | 84.03 | 79.67 | 76.05 | 80.99 | 84.90 | 89.51 | 83.96 |
| | | VCD | 83.23 | 76.73 | 85.05 | 82.83 | 80.16 | 83.56 | 85.21 | 92.05 | 84.21 |
| | | M3ID | 84.20 | 78.00 | 85.77 | 83.07 | 80.06 | 83.87 | 85.69 | 93.11 | 84.67 |
| | | RITUAL | 86.10 | 80.30 | 87.31 | 84.87 | 82.52 | **85.39** | **86.13** | 93.78 | 84.81 |
| | | **Ours** | **87.40** | **83.51** | **88.09** | **85.40** | **85.64** | 85.12 | 85.95 | **94.22** | **85.08** |
| | Popular | Regular | 73.47 | 66.83 | 77.84 | 73.33 | 68.72 | 76.26 | 81.33 | 83.38 | 80.74 |
| | | VCD | 72.37 | 65.27 | 77.58 | 76.13 | 71.10 | **78.68** | 81.97 | 82.82 | 81.73 |
| | | M3ID | 73.87 | 66.70 | 78.49 | 75.17 | 69.94 | 78.04 | **82.13** | 84.58 | 81.48 |
| | | RITUAL | 74.80 | 67.50 | 79.15 | 74.50 | 69.17 | 77.61 | 81.13 | 85.48 | 81.03 |
| | | **Ours** | **78.10** | **71.56** | **80.98** | **76.90** | **73.89** | 78.27 | 82.10 | **86.39** | **81.85** |
| | Adversarial | Regular | 68.60 | 62.43 | 74.84 | 68.60 | 63.94 | 73.10 | 79.03 | 80.43 | 78.54 |
| | | VCD | 68.83 | 62.26 | 75.43 | 71.00 | 65.75 | 75.14 | 80.87 | 81.07 | 80.80 |
| | | M3ID | 68.67 | 62.16 | 75.28 | 71.17 | 65.79 | **75.36** | 81.03 | 82.93 | **80.94** |
| | | RITUAL | 68.23 | 61.75 | 75.10 | 70.17 | 64.76 | 74.78 | 81.07 | 83.29 | 80.41 |
| | | **Ours** | **74.07** | **67.42** | **78.22** | **73.63** | **70.08** | 75.11 | **81.13** | **84.18** | 80.57 |

## 4.2 RESULTS AND DISCUSSIONS

**Results on POPE**. In Table 1, we compare the performance of our method against other baselines on the POPE benchmark under three different negative sampling settings, across three datasets. As shown in the table, our method consistently outperforms other decoding methods on three LVLMs, achieving state-of-the-art accuracies across all settings, with improvements of up to 5.24% in accuracy, 6.33% in precision, and 2.79% in F1 score compared to the second-best approach. This suggests that incorporating a generative reference enables the LVLMs to perceive more fine-grained visual details, thereby effectively addressing object hallucinations. Moreover, while most decoding methods tend to be overconfident in their responses, the self-correcting decoding mechanism in our method makes it more conservative in responding `Yes`, as evidenced by significantly higher precision across all settings. This highlights its enhanced performance in filtering out false positives and suppressing misinformation.

Table 2: **Results on CHAIR (Rohrbach et al., 2018) benchmark.** We limit the maximum number of new tokens to 64. Lower (↓) CHAIR$_S$, CHAIR$_I$ and higher (↑) recall and length indicate better performance. The best results in each setting are **bolded**, and the second-best are underlined.

| Method | LLaVA-1.5 | | | | InstructBLIP | | | |
|---|---|---|---|---|---|---|---|---|
| | CHAIR$_S$ ↓ | CHAIR$_I$ ↓ | Recall ↑ | Length ↑ | CHAIR$_S$ ↓ | CHAIR$_I$ ↓ | Recall ↑ | Length ↑ |
| Regular | 26.2 | 9.4 | 58.5 | 53.4 | 31.2 | 11.1 | 59.0 | 53.6 |
| VCD | 24.4 | 7.9 | 63.3 | 54.2 | 30.0 | 10.1 | 61.8 | 54.2 |
| M3ID | 21.4 | 6.3 | **64.4** | 53.5 | 30.8 | 10.4 | 62.6 | 53.4 |
| RITUAL | 22.4 | 6.9 | 63.0 | **54.9** | 26.6 | 8.9 | 63.4 | 55.3 |
| Woodpecker | 24.9 | 7.5 | 60.8 | 49.7 | 31.2 | 10.8 | 62.3 | 51.3 |
| HALC | 21.7 | 7.1 | 63.4 | 53.4 | 24.5 | 8.0 | 63.8 | 55.1 |
| **Ours** | **18.4** | **6.1** | 62.7 | 54.1 | **24.0** | **7.7** | **67.2** | **55.5** |

Table 3: **Results on MME-Hallucination (Fu et al., 2023) and MMBench (Liu et al., 2024d) benchmark.** We report the average MME scores along with the standard deviation across three random seeds for each subset. We also report the overall accuracy achieved by the different methods on the MMBench benchmark in the final column. Higher scores (↑) indicate better performance. The best results are **bolded**, and the second-best are underlined.

| Method | Object-level | | Attribute-level | | MME Score ↑ | MMBench ↑ |
|---|---|---|---|---|---|---|
| | Existence ↑ | Count ↑ | Position ↑ | Color ↑ | | |
| Regular | 173.75 $(\pm 4.79)$ | 121.67 $(\pm 12.47)$ | 117.92 $(\pm 3.69)$ | 149.17 $(\pm 7.51)$ | 562.50 $(\pm 3.96)$ | 64.1 |
| DoLa | 176.67 $(\pm 2.89)$ | 113.33 $(\pm 10.41)$ | 90.55 $(\pm 8.22)$ | 141.67 $(\pm 7.64)$ | 522.22 $(\pm 16.78)$ | 63.8 |
| OPERA | 183.33 $(\pm 6.45)$ | 137.22 $(\pm 6.31)$ | 122.78 $(\pm 2.55)$ | 155.00 $(\pm 5.00)$ | 598.33 $(\pm 10.41)$ | 64.4 |
| VCD | 186.67 $(\pm 5.77)$ | 125.56 $(\pm 3.47)$ | 128.89 $(\pm 6.73)$ | 139.45 $(\pm 12.51)$ | 580.56 $(\pm 15.13)$ | 64.6 |
| M3ID | 186.67 $(\pm 5.77)$ | 128.33 $(\pm 10.41)$ | 131.67 $(\pm 5.00)$ | 151.67 $(\pm 20.88)$ | 598.11 $(\pm 20.35)$ | 64.4 |
| RITUAL | 187.50 $(\pm 2.89)$ | 139.58 $(\pm 7.64)$ | 125.00 $(\pm 10.27)$ | 164.17 $(\pm 6.87)$ | 616.25 $(\pm 20.38)$ | 63.8 |
| Woodpecker | 187.50 $(\pm 2.89)$ | 125.00 $(\pm 0.00)$ | 126.66 $(\pm 2.89)$ | 149.17 $(\pm 17.34)$ | 588.33 $(\pm 10.00)$ | 64.0 |
| HALC | 183.33 $(\pm 0.00)$ | 133.33 $(\pm 5.77)$ | 107.92 $(\pm 3.69)$ | 155.00 $(\pm 5.00)$ | 579.58 $(\pm 9.07)$ | 64.2 |
| **Ours** | **188.33** $(\pm 2.89)$ | **150.00** $(\pm 7.64)$ | **133.89** $(\pm 3.85)$ | **172.22** $(\pm 3.47)$ | **644.44** $(\pm 9.18)$ | **65.5** |

Another notable finding is that our method shows significantly improved performance in the *popular* and *adversarial* settings, which are more challenging than the *random* setting. In the *popular* and *adversarial* settings, non-existent negative objects frequently appear and co-occur with other objects (Li et al., 2023d), making them more susceptible to hallucination by LVLMs, as evidenced by the varying degrees of performance degradation across all baselines. However, our method exhibits a lower performance drop compared to other baselines, demonstrating its effectiveness in addressing hallucinations arising from object co-occurrence.

**Results on CHAIR**. We also compare the performance of our methods and other state-of-the-art methods in the open-ended captioning task and report the CHAIR scores, recall, and the average length of responses in Table 2, Table C1, and Table C2. The results, evaluated across two different LVLMs, consistently demonstrate performance improvements achieved by our method over the compared approaches. Specifically, our method outperforms the second-best approach by 3.0% and 2.6% on the CHAIR$_S$ metric, while also enhancing the detailedness of generated responses compared to regular decoding, as indicated by the higher recall and increased response length. These results demonstrate that by incorporating generative feedback into the decoding process of LVLMs, our method effectively mitigates object hallucinations in open-ended captioning tasks.

**Results on MME-Hallucination and MMBench**. Beyond object hallucinations, we further compare the performance of our method with other approaches using the more comprehensive MME-Hallucination benchmark, which includes both object-level and attribute-level hallucinations. The results in Table 3 and Table C3 demonstrate that our method significantly outperforms the compared methods, with substantial margins in the total score metric (*e.g.*, +18.19 on LLaVA-1.5 and +21.11 on InstructBLIP) and consistently superior performance across various evaluation settings, achieving the best results in 6 out of 8 settings. Moreover, our method shows notable improvements on the attribute-level *color* subset, which is particularly challenging as it requires models to accurately capture subtle attribute information. This further illustrates the effectiveness of our approach in addressing a wide range of hallucinations, both at the object existence level and in finer-grained attribute recognition. Additionally, our proposed DeGF enhances the general multi-modal understanding capabilities of LVLMs, as evidenced by its superior performance on the MMBench benchmark.

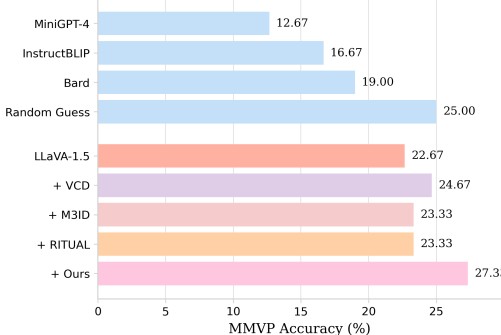

Figure 4: **Results on MMVP (Tong et al., 2024)**. We apply our approach to LLaVA-1.5 (Liu et al., 2024c) and compare its performance against other hallucination mitigation methods.

Table 4: **GPT-4V-aided evaluation on LLaVA-Bench**. Higher accuracy and detailedness (↑) indicate better performance. The evaluation is performed on LLaVA-1.5 (Liu et al., 2024c).

| Method | LLaVA-1.5 | | InstructBLIP | |
|---|---|---|---|---|
| | Acc. ↑ | Det. ↑ | Acc. ↑ | Det. ↑ |
| Regular | 2.88 | 3.29 | 3.42 | 3.96 |
| **Ours** | **4.29** | **4.54** | **4.38** | **4.79** |
| VCD | 3.62 | 3.83 | 3.71 | 4.21 |
| **Ours** | **4.04** | **4.38** | **4.17** | **4.58** |
| M3ID | 3.88 | 4.08 | 4.00 | 4.33 |
| **Ours** | **4.04** | **4.29** | **4.08** | **4.50** |

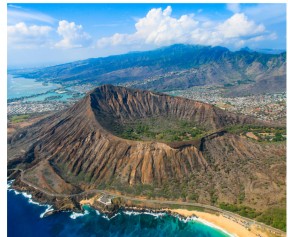

**Visual Input:**

**Query:** Describe this photo in detail.

[Regular Decoding]  GPT-4V-Aided Evaluation: Accuracy: 4 Detailedness: 5

This image portrays a strategically placed tropical hill, punctured with a road branching out through it, giving the appearance of a mountain. The island below the mountain appears to have beautiful blue waters, making for a picturesque view. On the East coast of the bay, there are several houses and buildings, some situated on the beach near the water. The overall skyline captures a cohesive and captivating island landscape. Take a closer look at the cloudy sky, which creates a serene and natural atmosphere for the charming scene.

[Our Approach]  GPT-4V-Aided Evaluation: Accuracy: 7 Detailedness: 7

The image displays a bird's eye view of an island with a large mountain in the center. The mountain is steep and appears to be a volcano, with greenery surrounding it. The island is located near the ocean, and it has a road running along the bottom of the mountain. The surrounding area is filled with houses, indicating that the island is inhabited. The island is surrounded by mountains, providing a picturesque and serene landscape. The combination of the ocean, the mountain, and the houses creates a unique and beautiful environment.

Figure 5: **Case study on the LLaVA-Bench benchmark**. We compare the responses generated by regular decoding and our method using LLaVA-1.5. GPT-4V-aided evaluation results are also provided alongside the responses. Hallucinated and accurate content is highlighted in red and green.

**Results on MMVP**. We conduct experiments on the MMVP benchmark to assess the fine-grained visual recognition capabilities of LVLMs. As shown in Figure 4, applying our self-correcting decoding approach to LLaVA-1.5 significantly improves performance from 22.67% to 27.33%. Our approach also demonstrates notable advantages over other hallucination mitigation baselines, further showcasing its superiority in handling nuanced visual recognition tasks. These results suggest that our approach significantly enhances the model's capacity to discern and correctly interpret fine-grained distinctions between images with similar appearances but different contents. By integrating generative feedback, our approach effectively reduces misinterpretations and improves the precision of visual recognition tasks, contributing to more reliable and accurate performance in complex scenarios.

**Results on LLaVA-Bench**. In Figure 5, we present a case study on LLaVA-Bench comparing our method's response with the response generated by regular decoding using the LLaVA-1.5 model. Specifically, regular decoding often leads to hallucinated or inaccurate content, such as describing "`the island below the mountain`". Besides, the response generated by regular decoding tends to focus on elements like the "`cloudy sky`" and "`cohesive and captivating island landscape`" without providing specific information about the central features of the image. In contrast, our response is more detailed, mentioning the volcano, the road, the surrounding greenery, and the inhabited areas, which gives a clearer understanding of the image's content. The GPT-4V-aided evaluation shown in Table 4 further confirms that our method enhances both the accuracy and detailedness of the generated response, outperforming other hallucination mitigation approaches such as VCD and M3ID. Due to the page limit, please refer to Section D of the Appendix for more case studies.

## 4.3 ABLATION STUDIES

**Analysis of Distance Threshold** $\gamma$. In Section 3.3, we introduce a distance threshold $\gamma$ to determine the appropriate decoding algorithm for each generated token. Table 5 presents an analysis of our method's performance with various values of $\gamma$ across three benchmarks. For simplicity, we report

Table 5: **Sensitivity analysis of distance threshold** $\gamma$. We present the performance of our approach, based on the LLaVA-1.5 backbone, across three benchmarks for varying values of $\gamma$.

| Values of $\gamma$ | POPE Acc. | CHAIR$_S$ | CHAIR$_I$ | MME Score |
|---|---|---|---|---|
| $\gamma = 0$ | 87.93 | 21.0 | 7.2 | 622.50 |
| $\gamma = 0.01$ | 88.07 | 21.0 | 6.8 | 632.22 |
| $\gamma = 0.05$ | 88.67 | 19.4 | 6.3 | 637.50 |
| $\gamma = 0.1$ | **89.03** | **18.4** | **6.1** | 644.44 |
| $\gamma = 0.5$ | 88.73 | 19.8 | 6.4 | **646.67** |
| $\gamma = 1$ | 88.43 | 21.6 | 6.6 | 638.33 |

Table 6: **Effects of different generative models.** We report the performance of different variants of our method, utilizing various stable diffusion models, on the LLaVA-1.5 backbone.

| Models | POPE Acc. | CHAIR$_S$ | CHAIR$_I$ | MME Score |
|---|---|---|---|---|
| Regular | 83.13 | 26.2 | 9.4 | 562.50 |
| SD-v1.1 | 88.37 | 19.3 | 6.5 | 638.33 |
| SD-v1.5 | **89.03** | 18.4 | 6.1 | 644.44 |
| SD-v2.1 | 88.70 | 18.8 | 6.7 | 632.22 |
| SD-XL-v0.9 | 88.87 | 18.6 | 6.1 | 642.50 |
| SD-XL-v1.0 | 88.60 | **17.9** | **5.8** | **648.33** |

the performance on the MS-COCO dataset with *random* setting for all POPE results in the ablation studies. Notably, when $\gamma$ is set to either 0 or 1—corresponding to the exclusive use of contrastive or complementary decoding for all tokens—the performance exhibits a significant decline, by 0.6% and 1.1% in POPE accuracy, respectively. Moreover, our default setting of $\gamma = 0.1$ achieves the optimal performance in 3 out of 4 evaluated metrics. Additional sensitivity analyses for other hyperparameters are provided in Section C of the Appendix.

**Effects of Different Generative Models**. Table 6 presents the performance of various variants of our method that incorporate different generative models (*i.e.*, different versions of Stable Diffusion) while using the same LLaVA-1.5 backbone. The results indicate that the effectiveness of our DeGF is robust to the choice of generative models, as performance remains largely unaffected by the specific model used, and all variants demonstrate consistent improvements over the original regular decoding approach. Although utilizing SD-XL-v1.0 (Podell et al., 2024) yields slightly better performance, we opt for SD-v1.5 as the default due to its faster image generation speed (3.8 s/image *vs*. 11.3 s/image).

## 4.4 Efficiency Comparison

In Table 7, we compare the efficiency of our approach with other methods on the CHAIR benchmark using the LLaVA-1.5 model, with the maximum token length set to 128. Our approach involves two queries and incorporates a text-to-image generative model to mitigate hallucinations, resulting in a $4.04\times$ increase in latency and a $1.21\times$ increase in GPU memory usage. Specifically, our method consists of three stages: initial response generation, image generation, and response self-correction, which take

Table 7: **Efficiency comparison**. For each method, we present the average inference latency per instance and peak GPU memory. Experiments are conducted on a single RTX A6000 Ada GPU.

| Method | Avg. Latency ↓ | GPU Memory ↓ | CHAIR$_S$ ↓ |
|---|---|---|---|
| Regular | 3.44 s $_{(\times 1.00)}$ | 15778 MB $_{(\times 1.00)}$ | 55.0 |
| VCD | 6.91 s $_{(\times 2.01)}$ | 16634 MB $_{(\times 1.05)}$ | 54.4 |
| OPERA | 24.70 s $_{(\times 7.18)}$ | 22706 MB $_{(\times 1.44)}$ | 52.6 |
| Woodpecker | 10.68 s $_{(\times 3.10)}$ | 22199 MB $_{(\times 1.41)}$ | 57.6 |
| HALC | 22.61 s $_{(\times 6.51)}$ | 23084 MB $_{(\times 1.46)}$ | 51.0 |
| **Ours** | 13.89 s $_{(\times 4.04)}$ | 19119 MB $_{(\times 1.21)}$ | 48.8 |

an average of 3.4 seconds, 3.8 seconds, and 6.6 seconds per instance, respectively. Compared to other approaches, while our method is slower than regular decoding and contrastive decoding-based methods, it demonstrates efficiency advantages over OPERA and HALC. Note that our approach also achieves the lowest hallucination rates among all compared methods. In Appendix C.9, we discuss several strategies to accelerate our approach, such as limiting the length of the initial response and reducing the number of inference steps in the diffusion process.

## 5 Conclusion

In this work, we present self-correcting Decoding with Generative Feedback (DeGF), a novel training-free approach that leverages feedback from text-to-image generative models to recursively improve the accuracy of generated responses. Specifically, we generate a new image based on the initial response given by LVLMs, which serves as a visual reference and provides token-level feedback for mitigating hallucinations. Building on this, we propose a corresponding self-correcting decoding algorithm that measures the discrepancy between next-token predictions conditioned on the original and generated images, selecting either contrastive or complementary decoding to reduce the likelihood of hallucinatory responses. Extensive experimental results across six benchmarks demonstrate that our DeGF consistently outperforms state-of-the-art methods in mitigating hallucinations in LVLMs.

## ACKNOWLEDGEMENTS

This work has been funded in part by the Army Research Laboratory (ARL) award W911NF-23-2-0007 and W911QX-24-F-0049, DARPA award FA8750-23-2-1015, and ONR award N00014-23-1-2840. MM and LPM are partially supported by Meta and National Institutes of Health awards R01MH125740, R01MH132225, and R21MH130767. RS is supported in part by the ONR grant N00014-23-1-2368.

## ETHICS STATEMENT

Our work focuses on enhancing the reliability of current large vision-language models. Our research does not involve human subjects, sensitive data, or any practices that pose privacy or security concerns. We also discuss the broader ethical and societal implications of this work in Appendix A.

## REPRODUCIBILITY STATEMENT

The large vision-language models utilized in our experiments, such as LLaVA and InstructBLIP, are open-source and publicly available. We have detailed our experimental setup, including hyperparameter configurations, prompts, and other key design choices, in Section 4 of the main paper and Section B of the Appendix to ensure reproducibility. Code is publicly available at https://github.com/zhangce01/DeGF.

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

# Self-Correcting Decoding with Generative Feedback for Mitigating Hallucinations in Large Vision-Language Models

# Appendix

In this supplementary document, we provide additional details and experimental results to enhance understanding and insights into our method. This supplementary document is organized as follows:

- The limitations and broader impacts of this work are discussed in Section A.
- Additional experimental details, including further implementation details, descriptions of other implemented baselines, and license information for the utilized code and datasets, are provided in Section B.
- Additional experimental results are presented in Section C.
- More case studies and GPT-4V-aided evaluations are provided in Section D.
- Potential directions for future work are discussed in Section E.

## A  Limitations and Broader Impacts

**Limitations**. Although our method effectively mitigates hallucinations in LVLMs, it relies on pre-trained text-to-image generative models, which introduces additional computational complexity. The process of generating images also adds time, potentially slowing down LVLM response generation and making it less suitable for real-time applications. However, our method is training-free, reducing the overhead typically associated with fine-tuning large models, and offering broader applicability across various tasks. Moreover, the use of generative feedback improves the model's ability to verify and correct responses, particularly in complex scenarios. Thus, while the computational trade-offs may limit real-time performance, our method excels in settings where accuracy and reliability are prioritized over speed. We also hope that advances in efficient diffusion-based models will improve the feasibility of our approach in real-world applications in the future.

**Broader Impacts**. In this work, our goal is to develop more reliable large vision-language models (LVLMs) by incorporating feedback from generative models. By using this feedback mechanism, we aim to address a critical issue faced by current multi-modal models: hallucinations, where models produce responses that are inconsistent with the visual input. Hallucinations not only degrade model performance but also pose risks in real-world applications by generating inaccurate or misleading information. Our approach leverages the strengths of generative models to detect and mitigate these hallucinations, improving the overall accuracy and reliability of LVLMs. In doing so, we contribute to enhancing trustworthiness and reducing the spread of misinformation in systems that rely on multi-modal AI, making them safer and more effective for a wide range of applications.

## B  More Experimental Details

### B.1  Benchmarks and Metrics

We conduct extensive experiments on the following benchmarks:

- **POPE** (Li et al., 2023d) is a widely used benchmark for assessing object hallucinations in LVLMs. It tests the models with yes-or-no questions regarding the presence of specific objects, such as, "Is there a {object} in the image?" The benchmark draws data from three existing datasets: MSCOCO (Lin et al., 2014), A-OKVQA (Schwenk et al., 2022), and GQA (Hudson & Manning, 2019), and comprises three distinct subsets—*random*, *popular*, and *adversarial*—based on how the negative samples are generated. For each dataset setting, the benchmark provides 6 questions per image, resulting in 3,000 test instances. We evaluate the performance of different methods using four metrics: accuracy, precision, recall, and F1 score.

- **CHAIR** (**Rohrbach et al., 2018**) evaluates object hallucinations in open-ended captioning tasks. It prompts the LVLMs to describe specific images selected from a random sample of 500 images from the MSCOCO validation set and assesses performance based on two metrics:

$$\text{CHAIR}_I = \frac{\text{\# hallucinated objects}}{\text{\# all objects mentioned}}, \quad \text{CHAIR}_S = \frac{\text{\# sentences with hallucinated object}}{\text{\# all sentences}}. \quad (6)$$

Additionally, we assess the recall and the average length of the generated responses.

- **MME-Hallucination** (**Fu et al., 2023**) is a comprehensive benchmark for LVLMs consisting of four subsets: *existence* and *count* for object-level hallucinations, and *position* and *color* for attribute-level hallucinations. Each subset includes 30 images and 60 questions, with two questions per image. Similar to POPE (Li et al., 2023d), these questions are structured as yes-or-no queries, and performance is assessed based on binary accuracy. Following the official implementation, the reported score is calculated by combining accuracy and accuracy+, where accuracy is based on individual questions, and accuracy+ is based on images where both questions are answered correctly.

- **MMBench** (**Liu et al., 2024d**) is a comprehensive evaluation benchmark designed to assess the multimodal understanding and reasoning capabilities of AI models. It focuses on tasks requiring the integration of visual and textual information, testing a model's ability to handle diverse, real-world scenarios. In particular, MMBench employs a hierarchical ability taxonomy, designating Perception and Reasoning as Level-1 (L-1) abilities. It further refines the taxonomy by incorporating more detailed ability dimensions, organizing them into six Level-2 (L-2) and twenty Level-3 (L-3) dimensions.

- **MMVP** (**Tong et al., 2024**) collects CLIP-blind pairs and evaluates the fine-grained visual recognition capabilities of LVLMs. It consists of 150 image pairs, each accompanied by a binary-option question. Each image is queried independently, and for a given pair, the LVLM's response is considered correct only if both associated questions are answered accurately.

- **LLaVA-Bench**[3] provides 24 images featuring complex scenes, memes, paintings, and sketches, along with 60 challenging questions. We select examples from this dataset to provide qualitative comparisons between the responses generated by different decoding methods. We also follow Yin et al. (2023) to evaluate the accuracy and detailedness of generated responses of different methods using the advanced LVLM, GPT-4V[4].

### B.2 More Implementation Details

In our experiments, we adhere to the default query format for the input data used in both LLaVA-1.5 (Liu et al., 2023) and InstructBLIP (Dai et al., 2023). Additionally, we set $\alpha_1 = 3$, $\alpha_2 = 1$, and $\gamma = 0.1$ by default in our decoding process. We follow VCD (Leng et al., 2024) to implement adaptive plausibility constraints (Li et al., 2023c):

$$p_\theta(y_t) = 0, \quad \text{if } y_t \notin \mathcal{V}(y_{<t})$$
$$\text{where } \mathcal{V}(y_{<t}) = \{y_t \in \mathcal{V} : p_\theta(y_t|v, \mathbf{x}, \mathbf{y}_{<t}) \geq \beta \max_w p_\theta(w|v, \mathbf{x}, \mathbf{y}_{<t})\} \quad (7)$$

Here, $\mathcal{V}$ is the whole vocabulary of LVLM, and hyperparameter $\beta \in [0, 1]$ controls the truncation of the next token distribution. A larger $\beta$ indicates more aggressive truncation, keeping only the high-probability tokens. In our implementation, we set the logits for $y_t \notin \mathcal{V}(y_{<t})$ to $-\infty$. By default, we set $\beta = 0.1$ in the open-ended CHAIR benchmark and $\beta = 0.25$ for other tasks. All experiments are conducted on a single 48GB NVIDIA RTX 6000 Ada GPU.

Recall that in our method, we use a text-to-image generative model to reverse the image-to-text response generation process by producing a new image from the initial response. To ensure the new image is both high-quality and relevant, we aim to generate specific descriptions for the given visual content. Thus, we slightly modify the initial query prompt for each evaluated benchmark:

- **POPE** (**Li et al., 2023d**), **MME-Hallucination** (**Fu et al., 2023**), **and MMVP** (**Tong et al., 2024**). In POPE, MME-Hallucination, and MMVP benchmarks, models are tested with yes-or-no/binary selection questions, such as, "Is there a {object} in the image?" To obtain more detailed explanations and descriptions of the original image, we modify the prompt by adding, "Briefly

---

[3]`https://huggingface.co/datasets/liuhaotian/llava-bench-in-the-wild`.
[4]`https://openai.com/index/gpt-4v-system-card`.

describe relevant details." This encourages the model to provide not only a yes-or-no answer but also additional visual information.

- **CHAIR (Rohrbach et al., 2018)**. For the CHAIR benchmark, we retain the original prompt, "Please describe this image in detail." as it effectively prompts the model to provide comprehensive visual details from the original image.

Note that for the second query, where both the original and generated images are used as input, we apply the original prompt to ensure a fair comparison.

### B.3 DETAILS OF OTHER BASELINES

In this work, we mainly compare the performance of our DeGF with three state-of-the-art approaches: VCD (Leng et al., 2024), M3ID (Favero et al., 2024), and RITUAL (Woo et al., 2024). The method and implementation details for these approaches are provided below:

- **VCD (Leng et al., 2024)** contrasts output distributions derived from original and distorted visual inputs. Specifically, given a textual query $x$ and a visual input $v$, the model generates two distinct output distributions: one conditioned on the original $v$ and the other on the distorted visual input $v'$, which is derived by applying pre-defined distortions (i.e., Gaussian noise mask) to $v$. Then, a new contrastive probability distribution is computed by:

$$p_{vcd}(y_t) = \text{Softmax}\left[(1+\alpha)f_\theta(y|v, \mathbf{x}, \mathbf{y}_{<t}) - \alpha f_\theta(y|v', \mathbf{x}, \mathbf{y}_{<t})\right]. \tag{8}$$

  In our implementation, we follow the default setting in VCD (Leng et al., 2024) and set $\alpha = 1$ for reproduction. To generate $v'$, we use a total of 500 noise steps.

- **M3ID (Favero et al., 2024)** contrasts output distributions derived from original visual inputs and pure text inputs without visual information. The final probability distribution is

$$p_{m3id}(y_t) = \text{Softmax}\left[f_\theta(y|v, \mathbf{x}, \mathbf{y}_{<t}) + \frac{1 - e^{-\lambda t}}{e^{-\lambda t}}\left(f_\theta(y|v, \mathbf{x}, \mathbf{y}_{<t}) - f_\theta(y|\mathbf{x}, \mathbf{y}_{<t})\right)\right]. \tag{9}$$

  Similarly, we follow their recommended best practice and set the hyperparameter $\lambda$, which balances the conditioned model and unconditioned model, to $0.02$.

- **RITUAL (Woo et al., 2024)** applies common image transformations (*e.g.*, crop, flip, color jitter, *etc.*) to the original visual input $v$, This results in a transformed version of the visual input, $v^{(T)}$. Then, RITUAL utilizes both the original and transformed images to generate the response and this dual-input approach significantly reduces the likelihood of hallucinatory outputs. The probability distribution is calculated as follows:

$$p_{ritual}(y_t) = \text{Softmax}\left[f_\theta(y|v, \mathbf{x}, \mathbf{y}_{<t}) + \kappa f_\theta\left(y|v^{(T)}, \mathbf{x}, \mathbf{y}_{<t}\right)\right]. \tag{10}$$

  Here, $\kappa$ is a balancing hyperparameter, adjusting the contribution of the transformed input relative to the original. We follow their official implementation to set $\kappa = 3$ in default.

### B.4 DATASET AND CODE LICENSING

**Datasets**. We list the known license information for the datasets below: POPE (Li et al., 2023d) and MMVP (Tong et al., 2024) benchmarks are licensed under MIT License. CHAIR (Rohrbach et al., 2018) is made available under the BSD 2-Clause License. LLaVA-Bench is available under Apache-2.0 License. MME-Hallucination (Fu et al., 2023) benchmark dataset is collected by Xiamen University for academic research only.

**Code**. In this work, we also use some code implementations from the existing codebase: LLaVA (Liu et al., 2023) and VCD (Leng et al., 2024) are licensed under the Apache-2.0 License. Instruct-BLIP (Dai et al., 2023) is under BSD-3-Clause License. RITUAL (Woo et al., 2024) is licensed under MIT License.

## C MORE EXPERIMENTAL RESULTS AND ANALYSIS

Table C1: **Results on CHAIR (Rohrbach et al., 2018) benchmark.** We limit the maximum number of new tokens to 128. Lower (↓) $CHAIR_S$, $CHAIR_I$ and higher (↑) recall and length indicate better performance. The best results in each setting are **bolded**, and the second-best are underlined.

| Method | LLaVA-1.5 | | | | InstructBLIP | | | |
|---|---|---|---|---|---|---|---|---|
| | $CHAIR_S$ ↓ | $CHAIR_I$ ↓ | Recall ↑ | Length ↑ | $CHAIR_S$ ↓ | $CHAIR_I$ ↓ | Recall ↑ | Length ↑ |
| Regular | 55.0 | 16.3 | 71.9 | **97.3** | 57.0 | 17.6 | 68.3 | **100.4** |
| VCD | 54.4 | 16.6 | 75.1 | 97.0 | 60.4 | 17.8 | **72.5** | 99.9 |
| M3ID | 56.6 | 15.7 | **76.8** | 94.5 | 62.2 | 18.1 | 71.9 | 99.8 |
| RITUAL | 49.6 | 14.8 | 74.7 | 96.2 | **48.4** | 14.5 | 72.2 | 100.0 |
| Woodpecker | 57.6 | 16.7 | 70.3 | 93.2 | 60.8 | 17.6 | 69.7 | 97.6 |
| HALC | 51.0 | 14.8 | 75.3 | 95.8 | 53.8 | 15.7 | 71.9 | 99.1 |
| **Ours** | **48.8** | **14.6** | 76.0 | 96.4 | 49.2 | **14.4** | 72.2 | 98.9 |

Table C2: **Results on CHAIR (Rohrbach et al., 2018) benchmark.** We limit the maximum number of new tokens to 256. Lower (↓) $CHAIR_S$, $CHAIR_I$ and higher (↑) recall and length indicate better performance. The best results in each setting are **bolded**, and the second-best are underlined.

| Method | LLaVA-1.5 | | | | InstructBLIP | | | |
|---|---|---|---|---|---|---|---|---|
| | $CHAIR_S$ ↓ | $CHAIR_I$ ↓ | Recall ↑ | Length ↑ | $CHAIR_S$ ↓ | $CHAIR_I$ ↓ | Recall ↑ | Length ↑ |
| Regular | 58.0 | 17.7 | 74.1 | **106.3** | 61.0 | 18.2 | 68.9 | **112.0** |
| VCD | 58.2 | 16.7 | 78.0 | 103.5 | 63.0 | 18.6 | **72.9** | 106.3 |
| M3ID | 56.8 | 16.1 | **80.7** | 98.2 | 65.8 | 19.9 | 72.4 | 102.7 |
| RITUAL | 51.0 | 15.1 | 76.0 | 100.9 | 50.4 | 15.3 | 72.0 | 102.0 |
| **Ours** | **49.8** | **14.7** | 77.2 | 103.3 | **49.8** | **15.1** | 72.3 | 103.3 |

Table C3: **Results on MME-Hallucination (Fu et al., 2023) benchmark.** We report the average MME scores along with the standard deviation across three random seeds for each subset. We also report the total scores achieved by the different methods across all four subsets in the final column. Higher scores (↑) indicate better performance. The best results are **bolded**, and the second-best are underlined.

| Model | Method | Object-level | | Attribute-level | | Total Score ↑ |
|---|---|---|---|---|---|---|
| | | Existence ↑ | Count ↑ | Position ↑ | Color ↑ | |
| LLaVA-1.5 | Regular | 173.75 (±4.79) | 121.67 (±12.47) | 117.92 (±3.69) | 149.17 (±7.51) | 562.50 (±3.96) |
| | DoLa | 176.67 (±2.89) | 113.33 (±10.41) | 90.55 (±8.22) | 141.67 (±7.64) | 522.22 (±16.78) |
| | OPERA | 183.33 (±6.45) | 137.22 (±6.31) | 122.78 (±2.55) | 155.00 (±5.00) | 598.33 (±10.41) |
| | VCD | 186.67 (±5.77) | 125.56 (±3.47) | 128.89 (±6.73) | 139.45 (±12.51) | 580.56 (±15.13) |
| | M3ID | 186.67 (±5.77) | 128.33 (±10.41) | 131.67 (±5.00) | 151.67 (±20.88) | 598.11 (±20.35) |
| | RITUAL | 187.50 (±2.89) | 139.58 (±7.64) | 125.00 (±10.27) | 164.17 (±6.87) | 616.25 (±20.38) |
| | **Ours** | **188.33** (±2.89) | **150.00** (±7.64) | **133.89** (±3.85) | **172.22** (±3.47) | **644.44** (±9.18) |
| InstructBLIP | Regular | 160.42 (±5.16) | 79.17 (±8.22) | 79.58 (±8.54) | 130.42 (±17.34) | 449.58 (±24.09) |
| | DoLa | 175.00 (±5.00) | 55.00 (±5.00) | 48.89 (±3.47) | 113.33 (±6.67) | 392.22 (±7.88) |
| | OPERA | 175.00 (±3.33) | 61.11 (±3.47) | 53.89 (±1.92) | 120.55 (±2.55) | 410.56 (±9.07) |
| | VCD | 158.89 (±5.85) | **91.67** (±18.34) | 66.11 (±9.76) | 121.67 (±12.58) | 438.33 (±16.07) |
| | M3ID | 160.00 (±5.00) | 87.22 (±22.63) | 69.44 (±9.18) | 125.00 (±7.64) | 441.67 (±17.32) |
| | RITUAL | 182.50 (±6.45) | 74.58 (±5.99) | 67.08 (±10.31) | 139.17 (±0.96) | 463.33 (±12.40) |
| | **Ours** | **186.67** (±2.89) | 89.44 (±8.22) | 58.33 (±4.41) | **150.00** (±1.89) | **484.44** (±11.34) |
| Qwen-VL | Regular | 155.00 (±3.54) | 127.67 (±13.36) | 131.67 (±7.73) | 173.00 (±9.75) | 587.33 (±31.06) |
| | VCD | 156.00 (±6.52) | 131.00 (±6.19) | 128.00 (±3.61) | 181.67 (±5.14) | 596.67 (±11.61) |
| | M3ID | 178.33 (±2.89) | 143.33 (±2.89) | 150.00 (±2.89) | 175.00 (±5.00) | 646.66 (±8.50) |
| | RITUAL | 178.33 (±2.89) | 142.22 (±16.19) | **156.66** (±2.89) | 178.33 (±2.89) | 655.55 (±14.99) |
| | **Ours** | **180.00** (±0.00) | 148.89 (±6.74) | 155.00 (±7.64) | 178.33 (±2.89) | **662.22** (±4.37) |

## C.1 ADDITIONAL RESULTS ON CHAIR

In Table C1 and Table C2, we present performance comparisons on the CHAIR benchmark with maximum number of tokens set to 128 and 256. The results indicate that our approach also achieves competitive performance across two LVLMs in mitigating hallucinations during long-sequence generation scenarios.

Table C4: **Detailed results on MMBench benchmark**. Abbreviations adopted: LR for Logical Reasoning; AR for Attribute Reasoning; RR for Relation Reasoning; FP-S for Fine-grained Perception (Single Instance); FP-C for Fine-grained Perception (Cross Instance); CP for Coarse Perception. The best results are **bolded**.

| Method | LR | AR | RR | FP-S | FP-C | CP | Overall |
|--------|------|-------|-------|-------|-------|-------|---------|
| Regular | 30.51 | 71.36 | 52.17 | 67.58 | **58.74** | 76.35 | 64.09 |
| VCD | 30.51 | **73.37** | 53.04 | 67.92 | 57.34 | 77.03 | 64.60 |
| M3ID | 30.51 | 72.36 | 53.04 | 67.58 | 57.34 | **77.36** | 64.43 |
| RITUAL | 28.81 | 72.86 | 54.78 | 65.87 | 58.04 | 76.01 | 63.83 |
| **Ours** | **31.36** | 70.85 | **60.87** | **68.60** | **58.74** | **77.36** | **65.46** |

Table C5: **Detailed results on MMVet benchmark with regular sampling**. Abbreviations adopted: Rec for Recognition, OCR for Optical Character Recognition, Know for Knowledge, Gen for Language Generation, Spat for Spatial Awareness, Math for Mathematics. The best results are **bolded**, and the second best are underlined.

| Method | Rec | OCR | Know | Gen | Spat | Math | Total |
|--------|------|------|------|------|------|------|-------|
| Regular | 30.8 | 19.0 | 14.5 | 17.9 | 26.9 | **11.5** | 26.1 |
| VCD | 35.6 | 21.9 | 18.3 | 21.9 | 28.9 | 3.8 | 30.9 |
| M3ID | 35.0 | 19.7 | 18.8 | 19.0 | 26.0 | 7.7 | 29.9 |
| RITUAL | **36.3** | 20.6 | **19.5** | 21.1 | 24.7 | 7.7 | 30.6 |
| **Ours** | 35.9 | 27.2 | 19.2 | **22.4** | **30.4** | **11.5** | **33.0** |

## C.2 FULL RESULTS ON MME-HALLUCINATION

In Table C3, we present the full results on the MME-Hallucination benchmark across three LVLMs.

## C.3 FULL RESULTS ON MMBENCH

In Table C4, we present the overall performance on the MMBench benchmark, as well as the detailed performance across six Level-2 abilities: Logical Reasoning (LR), Attribute Reasoning (AR), Relation Reasoning (RR), Fine-grained Perception - Single Instance (FP-S), Fine-grained Perception - Cross Instance (FP-C), and Coarse Perception (CP). We follow VCD (Leng et al., 2024) to conduct experiments on the MMBench-`dev` set.

## C.4 RESULTS ON MM-VET

In Table C5 and Table C6, we present the overall performance on the MM-Vet (Yu et al., 2024) benchmark with random sampling decoding and greedy decoding strategies, respectively. We use LLaVA-1.5 as the LVLM backbone. From the results, we observed that our method consistently outperforms others on the MMVet benchmark. Notably, it significantly excels in the OCR, spatial awareness, and math subsets.

## C.5 RESULTS ON POPE USING GREEDY DECODING

In Table C7, we present performance comparisons on the POPE benchmark with random sampling from the MS-COCO dataset. The experiment is conducted using the LLaVA-1.5 backbone.

## C.6 EFFECTS OF $\alpha_1$ AND $\alpha_2$ IN SELF-CORRECTING DECODING

In Section 3, we present two decoding approaches: complementary decoding and contrastive decoding. We also introduce two balancing hyperparameters, $\alpha_1$ and $\alpha_2$, which control the relative influence of

Table C6: **Detailed results on MMVet benchmark with greedy decoding**. Abbreviations adopted: Rec for Recognition, OCR for Optical Character Recognition, Know for Knowledge, Gen for Language Generation, Spat for Spatial Awareness, Math for Mathematics. The best results are **bolded**, and the second best are underlined.

| Method | Rec | OCR | Know | Gen | Spat | Math | Total |
|---|---|---|---|---|---|---|---|
| Greedy | 37.0 | 22.6 | 17.5 | 20.2 | 24.9 | 7.7 | 31.8 |
| VCD | **38.2** | 22.8 | **22.5** | **24.6** | 25.1 | 3.8 | 33.4 |
| M3ID | 37.9 | **23.6** | 20.4 | 20.7 | 26.0 | 11.5 | 33.2 |
| RITUAL | 35.6 | 21.7 | 18.9 | 19.9 | 24.7 | 7.7 | 30.6 |
| **Ours** | 37.9 | **25.0** | 20.2 | 19.5 | **32.8** | **15.0** | **34.0** |

Table C7: **Results on POPE using greedy decoding**.

| Values | POPE | | | |
|---|---|---|---|---|
| | Acc. | Prec. | Rec. | F1 |
| Greedy | 87.73 | 88.19 | 87.13 | 87.66 |
| VCD | 87.47 | 86.64 | **88.60** | 87.61 |
| M3ID | 89.07 | 89.54 | 88.47 | 89.00 |
| RITUAL | 89.23 | 90.17 | 88.07 | 89.11 |
| **Ours** | **89.40** | **94.44** | 83.73 | **88.76** |

the original and generated images in next-token prediction. In Table C8 and Table C9, we analyze the effect of varying $\alpha_1$ or $\alpha_2$ while keeping all other hyperparameters at their default settings. The results indicate that our default choice of $\alpha_1 = 3$ and $\alpha_2 = 1$ consistently yields the best performance across two benchmarks. Moreover, compared to setting these hyperparameters to 0, which effectively reduces complementary/contrastive decoding to standard decoding, the performance improvements demonstrate that our proposed decoding approaches significantly contribute to the overall effectiveness of DeGF in mitigating hallucinations in LVLMs.

## C.7 EFFECT OF $\beta$ IN ADAPTIVE PLAUSIBILITY CONSTRAINT

We further conduct an ablation study on $\beta$ introduced in Equation (7), where we vary $\beta$ from 0 to 0.5 while keeping all other hyperparameters fixed. The results in Table C10 show that setting $\beta = 0$, which imposes no constraint, results in suboptimal performance across both benchmarks. Additionally, in the POPE benchmark, where LVLMs handle yes-or-no questions, a more aggressive truncation with $\beta = 0.25$ yields the best performance. In contrast, for the open-ended CHAIR benchmark, a lower value of $\beta = 0.1$ leads to the best results.

## C.8 SCALING UP THE LVLMS

We further extend our evaluation to larger-scale 13B variants of the LLaVA-1.5 model to assess the scalability of our approach. Table C11 compares our experimental results with other state-of-the-art approaches across all three subsets of the POPE benchmark using the 13B-sized LLaVA-1.5 model. We observe that scaling up the LLaVA-1.5 model does not alleviate the hallucination issues, as evidenced by the comparable performance of both the 7B and 13B models. Using the 13B-sized model, our DeGF consistently achieves improved performance across all subsets compared to other approaches, demonstrating its general effectiveness and scalability.

## C.9 SPEEDING UP OUR APPROACH

In this section, we propose two strategies to accelerate our approach: limiting the length of the initial response and reducing the number of inference steps in the diffusion process.

Table C8: **Sensitivity analysis of hyperparameter** $\alpha_1$. We present the performance of our approach, based on the LLaVA-1.5 backbone, across two benchmarks for varying values of $\alpha_1$. Note that we fix $\alpha_2 = 1$ in this experiment.

| Values | POPE | | | | CHAIR | |
|---|---|---|---|---|---|---|
| | Acc. | Prec. | Rec. | F1 | CHAIR$_S$ | CHAIR$_I$ |
| $\alpha_1 = 0$ | 87.50 | 86.71 | 87.49 | 87.10 | 22.8 | 7.6 |
| $\alpha_1 = 1$ | 87.97 | 87.28 | 87.34 | 87.31 | 20.6 | 6.9 |
| $\alpha_1 = 2$ | 88.90 | 89.39 | **87.75** | 88.56 | 19.4 | 6.3 |
| $\alpha_1 = 3$ | **89.03** | **91.20** | 86.40 | **88.74** | **18.4** | **6.1** |
| $\alpha_1 = 4$ | 88.67 | 90.56 | 85.28 | 87.84 | 22.6 | 8.1 |

Table C9: **Sensitivity analysis of hyperparameter** $\alpha_2$. We present the performance of our approach, based on the LLaVA-1.5 backbone, across two benchmarks for varying values of $\alpha_2$. Note that we fix $\alpha_1 = 3$ in this experiment.

| Values | POPE | | | | CHAIR | |
|---|---|---|---|---|---|---|
| | Acc. | Prec. | Rec. | F1 | CHAIR$_S$ | CHAIR$_I$ |
| $\alpha_2 = 0$ | 86.77 | 85.17 | 86.58 | 85.87 | 23.6 | 8.2 |
| $\alpha_2 = 1$ | **89.03** | **91.20** | 86.40 | **88.74** | **18.4** | **6.1** |
| $\alpha_2 = 2$ | 88.73 | 89.86 | **86.66** | 88.23 | 21.8 | 7.5 |
| $\alpha_2 = 3$ | 88.03 | 87.97 | 86.28 | 87.12 | 22.8 | 7.3 |
| $\alpha_2 = 4$ | 87.13 | 86.52 | 86.16 | 86.34 | 23.6 | 7.9 |

- **Reducing Diffusion Inference Steps**. By default, we set the number of diffusion inference steps to 50 to ensure high-quality image generation. To improve the response generation speed, we can reduce the number of diffusion steps. In Table C12, we report the performance on the CHAIR benchmark after reducing the diffusion inference steps in the model. By reducing the diffusion inference steps from 50 to 10, the average latency decreases by 2.85 seconds per instance, while the performance on CHAIR remains robust. This demonstrates that reducing the inference steps of the diffusion model is an effective way to speed up our approach.

- **Restricting Length of Initial Response**. Our method involves two queries to the LVLM for self-correcting decoding. To enhance efficiency, we can limit the length of the initial response. In Table C13, we present the efficiency and CHAIR performance results after decreasing the maximum token limit for the initial response. We can see that reducing the maximum number of tokens in the initial response from 128 to 96 decreases the latency by 0.72 seconds per instance while maintaining competitive performance. However, further reductions result in performance degradation, as a shorter initial response fails to adequately cover the entire scene, limiting its ability to generate an image that effectively reflects and mitigates hallucinations.

Note that these two strategies are not conflicting; instead, they are complementary. Setting the diffusion steps to 10 and limiting the maximum number of tokens in the initial response to 96 further reduces the inference latency to 10.21 seconds per instance while maintaining robust performance.

## C.10 QUANTITATIVE ASSESSMENT OF GENERATED IMAGE QUALITY

Our approach incorporates a text-to-image generation model to mitigate hallucinations. We evaluate the quality of the generated images on all 4 subsets on the MME benchmark using CLIPScore (Hessel et al., 2021). Specifically, we utilize the CLIP backbone with ViT-B/32 backbone for our evaluation. We list the results in Table C14. As we can see from the table, our text-to-image generative model (specifically, SD-v1.5) achieves an average CLIPScore of over 30 across all subsets. For comparison,

Table C10: **Sensitivity analysis of hyperparameter** $\beta$. We present the performance of our approach, based on the LLaVA-1.5 backbone, across two benchmarks for varying values of $\beta$.

| Values | POPE | | | | CHAIR | |
|---|---|---|---|---|---|---|
| | Acc. | Prec. | Rec. | F1 | $CHAIR_S$ | $CHAIR_I$ |
| $\beta = 0$ | 87.17 | 87.45 | 85.30 | 86.36 | 21.2 | 7.1 |
| $\beta = 0.05$ | 88.27 | 89.85 | 86.12 | 87.95 | 19.1 | 6.3 |
| $\beta = 0.1$ | 88.33 | 89.04 | 86.04 | 87.52 | **18.4** | **6.1** |
| $\beta = 0.25$ | **89.03** | **91.20** | **86.40** | **88.74** | 19.3 | 6.5 |
| $\beta = 0.5$ | 87.80 | 88.79 | 85.48 | 87.10 | 20.2 | 6.9 |

Table C11: **Results on POPE (Li et al., 2023d) benchmark using 13B-sized LLaVA-1.5**. Higher (↑) accuracy, precision, recall, and F1 indicate better performance.

| Setup | Method | LLaVA-1.5 | | | |
|---|---|---|---|---|---|
| | | Acc. ↑ | Prec. ↑ | Rec. ↑ | F1 ↑ |
| **MS-COCO** — Random | Regular | 82.53 | 78.57 | 89.47 | 83.67 |
| | VCD | 84.80 | 80.67 | 91.53 | 85.76 |
| | M3ID | 85.37 | 81.30 | 91.87 | 86.26 |
| | RITUAL | 87.80 | 84.45 | **92.67** | **88.37** |
| | **Ours** | **88.40** | **88.14** | 88.61 | **88.37** |
| Popular | Regular | 80.53 | 76.17 | 88.87 | 82.03 |
| | VCD | 82.23 | 76.88 | 92.20 | 83.84 |
| | M3ID | 82.60 | 77.91 | 91.00 | 83.95 |
| | RITUAL | 84.07 | 79.00 | **92.80** | 85.35 |
| | **Ours** | **85.30** | **84.18** | 86.93 | **85.53** |
| Adversarial | Regular | 75.80 | 70.41 | 89.00 | 78.62 |
| | VCD | 77.33 | 71.44 | 91.07 | 80.07 |
| | M3ID | 77.43 | 71.65 | 90.80 | 80.09 |
| | RITUAL | 78.00 | 71.72 | **92.47** | 80.78 |
| | **Ours** | **81.43** | **78.61** | 87.04 | **82.61** |

the advanced DALL-E 3 model achieves a score of 32.0, while DALL-E 2 achieves 31.4.[5] These results highlight the capability of our model to generate high-quality images that closely align with the initial response.

# D  MORE CASE STUDIES

## D.1  DETAILS ABOUT GPT-4V-AIDED EVALUATION

Following VCD (Leng et al., 2024), we use GPT-4V to evaluate responses in open-ended generation scenarios, scoring them based on accuracy and detailedness. Leveraging the strong human-like capabilities of GPT-4V, it can detect incorrect colors, positions, and relationships, providing a comprehensive evaluation of the responses. Specifically, we apply the prompt provided in Table D15 to instruct GPT-4V to rate the two responses on a scale of 1 to 10 for both accuracy and detailedness:

- **Accuracy** measures the consistency between the responses/descriptions generated by the LVLMs and the given image. A lower score is assigned if GPT-4V detects any inconsistencies in the content of the responses.

- **Detailedness** evaluates the depth and specificity of the responses provided by the LVLMs. A higher score is awarded if the response includes comprehensive descriptions, captures fine-grained details

---

[5]These results are sourced from the technical report on DALL-E 3, available at: `https://cdn.openai.com/papers/dall-e-3.pdf`.

Table C12: **Effect of reducing diffusion inference steps.**

| Diff. Steps | Avg. Latency ↓ | CHAIR$_S$ ↓ | CHAIR$_I$ ↓ |
|---|---|---|---|
| 50 | 13.89 s | 48.8 | 14.6 |
| 30 | 12.56 s | 48.9 | 14.7 |
| 20 | 11.87 s | 49.2 | 14.8 |
| 10 | 11.04 s | 48.8 | 14.9 |

Table C13: **Effect of restricting the number of tokens in the initial response.**

| # Tokens | Avg. Latency ↓ | CHAIR$_S$ ↓ | CHAIR$_I$ ↓ |
|---|---|---|---|
| 128 | 13.89 s | 48.8 | 14.6 |
| 96 | 13.17 s | 48.8 | 14.9 |
| 64 | 12.20 s | 49.5 | 14.8 |
| 32 | 11.33 s | 51.2 | 14.9 |

Table C14: **CLIPScore evaluation across different MME subsets.**

| MME Subset | Existence | Count | Position | Color |
|---|---|---|---|---|
| **Avg. CLIPScore** | 31.34 | 30.69 | 30.09 | 31.69 |

of the image, and provides well-elaborated explanations. Conversely, a lower score is given if the response is vague or lacks sufficient detail.

## D.2 MORE QUALITATIVE RESULTS

In Figure D1 and Figure D2, we provide additional case studies on LLaVA-Bench to qualitatively demonstrate the effectiveness of our methods in mitigating hallucinations. We also included GPT-4V evaluations of accuracy and detailedness scores for each instance.

In Figure D3-D6, we provide qualitative evaluations of the images generated by the generative model, including both success and failure cases, across all four subsets of the MME benchmark to better understand the effectiveness of the generative models. Our results show that, despite occasional failure cases, the generative model consistently produces high-quality and realistic images that accurately visualize the initial response, providing effective self-feedback.

## E FUTURE WORK

In future work, we aim to extend the evaluation of our method to a broader range of LVLMs, such as Mini-GPT4 (Zhu et al., 2024) and mPLUG-Owl2 (Ye et al., 2024), as well as additional benchmarks, including R-Bench (Wu et al., 2024), which focuses on relation hallucination, and ROPE (Chen et al., 2024c), which addresses multiple-object hallucination. This expanded evaluation will allow us to more comprehensively assess the generalizability and effectiveness of our approach across diverse models and tasks.

Furthermore, we plan to investigate integrating generative feedback directly into the instruction tuning phase. This integration has the potential to eliminate the computational overhead associated with applying our method during inference, thereby significantly improving efficiency without compromising performance. By pursuing these directions, we hope to further enhance the practical applicability and scalability of our approach.

**Description:**
AI that scores image description accuracy and detailedness.

**Instructions:**
You are an AI designed to evaluate and score the performance of two AI assistants in describing a given image. Your primary focus is on the accuracy and detailedness of their descriptions. You will assess the accuracy by checking for hallucinations - any part of the description that is inconsistent with the image content. For detailedness, you will consider how rich the response is in necessary details, excluding any hallucinated parts. You will provide scores on a scale from 1 to 10 for each assistant separately, based on these criteria. After scoring, you will offer an explanation for your evaluation, ensuring it is free from bias and not influenced by the order of presentation of the responses.

Input format:

[Assistant 1]
{Response 1}
[End of Assistant 1]

[Assistant 2]
{Response 2}
[End of Assistant 2]

Output format:

Accuracy:
Scores of the two answers:
Reason:

Detailedness:
Scores of the two answers:
Reason:

Table D15: **GPT-4V-aided evaluation setup**. We present the prompt we provided to GPT-4V to evaluate the LVLM responses based on accuracy and detailedness.

Visual Input:

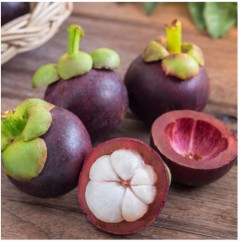

Query: Describe this photo in detail.

[Regular Decoding]                    GPT-4V-Aided Evaluation: Accuracy: 3 Detailedness: 4

The image features a table with four black fruit, possibly berries, cut in half to reveal their white interiors. The fruits are placed on the table, with one on the left side, one in the center, and two on the right side. In addition to the fruits, there are two bananas in the scene. One banana is located on the left side of the table, while the other is on the right side. The combination of fruits and bananas creates a colorful and appetizing display.

[Our Approach]                        GPT-4V-Aided Evaluation: Accuracy: 7 Detailedness: 6

The image displays an arrangement of black and green fruits on a wooden surface, possibly a wooden table. The fruits are the central focus, covering a large portion of the foreground. Four of these fruit cut in half can be seen, with some positioned closer to the viewer's perspective, making them the most noticeable elements of the scene. The other two fruits on the table are partially visible but appear to be obscured by the more central fruits, adding an interesting contrast between the visible and obscured elements in the image.

Visual Input:

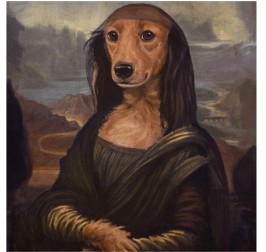

Query: Describe this photo in detail.

[Regular Decoding]                    GPT-4V-Aided Evaluation: Accuracy: 2 Detailedness: 3

The image is a painting of a dog wearing an Elizabethan style ruff or collar, giving it a "painted" appearance alongside Pinocchio. The dog is posing for the painting, as if it's commissioned. It's seated while holding a book in its paws. In the background of the painting, there is another large dog visible, perhaps meant to further emphasize the dog's artistic appearance. Around the scene, a handbag can be seen positioned near the top-right corner of the painting.

[Our Approach]                        GPT-4V-Aided Evaluation: Accuracy: 7 Detailedness: 6

The image features a dog wearing a hood and a gown, giving it an artistic and mysterious appearance. The dog is positioned in the center of the scene, and its attire appears to be inspired by the famous Mona Lisa painting. The hood and gown give the dog a regal and enigmatic look, reminiscent of the famous portrait of the Mona Lisa. The image is painted in a black and white style, further emphasizing the artistic nature of the scene.

Visual Input:

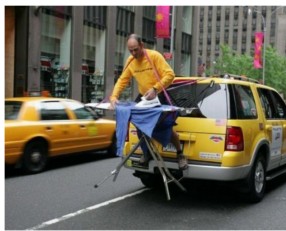

Query: Describe this photo in detail.

[Regular Decoding]                    GPT-4V-Aided Evaluation: Accuracy: 3 Detailedness: 4

The scene depicts a man sitting on a small folding chair, ironing a blue shirt on the back of a yellow city cab parked on a crowded city street. A red car is parked in front of the yellow cab, in close proximity to the overall scene. There are several other vehicles on the street as well, including a truck and a few cars nearer to the foreground. Among the parked cars, there is a teddy bear visible, adding a playful touch to the urban surroundings. The man seems to be catching a break by doing some duty in the midst of traffic.

[Our Approach]                        GPT-4V-Aided Evaluation: Accuracy: 4 Detailedness: 5

The image depicts a man riding on the back of a yellow taxi while ironing clothes. He is sitting on a portable ironing board, which is strapped to the taxi's back. The truck is driving down a busy street, with other vehicles such as cars and a bus visible in the scene. There are several plants placed along the street, adding some greenery to the urban environment. The man appears to be enjoying his unique mode of transportation while taking care of his laundry.

Visual Input:

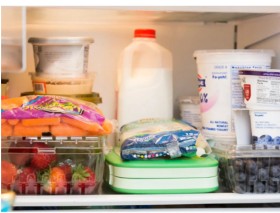

Query: Describe this photo in detail.

[Regular Decoding]                    GPT-4V-Aided Evaluation: Accuracy: 4 Detailedness: 4

The image features a well-stocked refrigerator filled with a variety of food items. On its shelves, there are several bottles, some filled and others empty, occupying varying positions and heights. In addition to the bottles, there are fresh fruits like grapes and strawberries displayed, and multiple carrots arranged in the bottom part of the fridge. The diverse array of items, including some fruits, vegetables, and beverages, showcase an organized and well-stocked refrigerator with a focus on healthy eating options.

[Our Approach]                        GPT-4V-Aided Evaluation: Accuracy: 6 Detailedness: 6

The image displays a well-stocked refrigerator filled with a variety of food items. There are several carrots placed in different compartments, with some in a plastic container and others in a green tray. The refrigerator also contains milk, yogurt, juice, and strawberries, all neatly arranged to create an organized and visually appealing display. In addition to the food items, there are a few bowls placed within the refrigerator, possibly for holding leftovers or additional ingredients. The refrigerator is filled with a diverse assortment of foods, showcasing a well-stocked and well-organized kitchen.

Figure D1: **Case studies on the LLaVA-Bench benchmark**. We compare the responses generated by regular decoding and our method using LLaVA-1.5. GPT-4V-aided evaluation results are also provided alongside the responses. Hallucinated and accurate content is highlighted in red and green.

**Visual Input:**

Sometimes I just look at pictures of the earth from space and I marvel at how beautiful it all is.

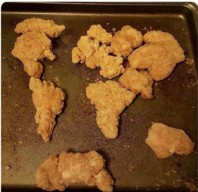

**Query:** Can you explain this meme in detail?

[Regular Decoding]                    GPT-4V-Aided Evaluation: Accuracy: 3 Detailedness: 4

In this image, there is a pan placed on the stovetop topped with multiple pieces of golden fried chicken. The cooked chicken pieces are scattered around the pan, creating a circle formation around it. The arrangement of the chicken pieces and their positioning over the pan give the impression of the Earth, as if the pieces of chicken are flying in the sky or traveling through space. This creative presentation gives the appearance of a global exposure or "I marveled at how beautiful it all is."

[Our Approach]                        GPT-4V-Aided Evaluation: Accuracy: 7 Detailedness: 6

The image features a fried chicken patty in the shape of a world map, placed on a metal baking sheet. The chicken patty is cut into several pieces, resembling different countries around the globe. The chicken pieces are arranged to represent various countries, with some pieces being larger and more prominent than others. The unique presentation of the chicken as a world map makes it an interesting and creative dish.

**Visual Input:**

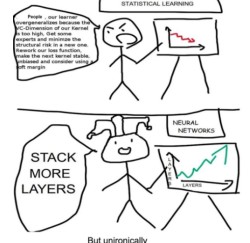

**Query:** Give a detailed description of this meme.

[Regular Decoding]                    GPT-4V-Aided Evaluation: Accuracy: 2 Detailedness: 3

The image is a comic strip featuring a man with a stack of heads on his chest, symbolizing a stack of layers, and several points on his face. He is talking about stacking layers. The comic is placed next to a smaller version of the same scene, and both scenes appear to be related to the topic of neural networks or layered systems. In addition to the man with the stack of heads, there is another pop character, John McClane himself, whose face is also incorporated into the scene, making a humorous connection to the main concept.

[Our Approach]                        GPT-4V-Aided Evaluation: Accuracy: 4 Detailedness: 5

The image consists of two comics depicting a person teaching statistics. In the first comic, the person is shown pointing to a graph, while in the second comic, they are pointing to a different graph. The person is explaining layers, stack more layers, and neural networks to the audience. The image is filled with various graphs and diagrams, which are used to illustrate the concepts being discussed. The person appears to be angry while teaching, emphasizing the importance of the topic. The comic is a mix of serious and humorous elements, making it an engaging and informative visual.

**Visual Input:**

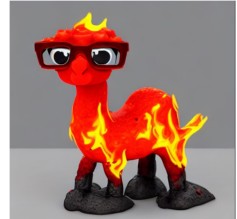

**Query:** This is the logo of LLaVA, Large Language and Vision Assistant, based on the LLaMA architecture. Please explain this logo in detail, and how do you think of its design.

[Regular Decoding]                    GPT-4V-Aided Evaluation: Accuracy: 2 Detailedness: 3

The LLaVA logo features a large eyeball, possibly a goofy looking one, with vibrant orange and red flames licking upward around its borders. The overall design appears to be quite intricate and abstract. The choice of colors and the unconventional design might be inspired by an amusing parody or a playful take on artificial intelligence and the classic HAL 9000 reference from 2001: A Space Odyssey. The stark contrast between the eerily composed "eye" and the flames gives the logo a unique aesthetic, sparking curiosity and attention for the viewer, while possibly conveying the concept of artificial intelligence. The air of humor or creative unsettlement may effectively communicate the underlying message and lure more people to appreciate the idea behind LLaVA.

[Our Approach]                        GPT-4V-Aided Evaluation: Accuracy: 8 Detailedness: 7

The LLaVA logo is a playful and unique design featuring a fire-breathing toy camel. The camel is standing on two black feet, with flames coming out of its mouth and legs. The glasses on the camel's face add a touch of whimsy to the design. The overall appearance of the logo is visually engaging and memorable, likely aiming to represent the innovative and creative nature of the LLaMA architecture. The design choice of using a camel as the central figure is an unconventional and eye-catching element, which can help the logo stand out and make a strong impression on viewers.

**Visual Input:**

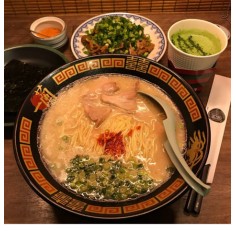

**Query:** Describe this photo in detail.

[Regular Decoding]                    GPT-4V-Aided Evaluation: Accuracy: 2 Detailedness: 3

The scene displays a delicious meal with a bowl of soup, a bowl of gyro, and another bowl containing greed. There's also a fried egg on rice in a bowl, offering a tempting mix of colors and textures. A spoon and a fork are placed near the soup, ready for use when enjoying this mouthwatering assortment. The various foods and bowls present an enticing feast likely appealing to those with a diverse appetite.

[Our Approach]                        GPT-4V-Aided Evaluation: Accuracy: 4 Detailedness: 5

The image features a dining table with a large bowl of delicious noodles, meat, and vegetables. The bowl is filled with various ingredients, such as onions, carrots, and green peppers, making it an appetizing meal. In addition to the main bowl, there are several smaller bowls placed on the table, containing different food items. A spoon is also visible on the table, likely for serving the noodle dish. The arrangement of the bowls and the spoon suggests that this is a meal shared among friends or family.

Figure D2: **Case studies on the LLaVA-Bench benchmark**. We compare the responses generated by regular decoding and our method using LLaVA-1.5. GPT-4V-aided evaluation results are also provided alongside the responses. Hallucinated and accurate content is highlighted in red and green.

Query: Is there a tie in this image? Describe relevant details about this.

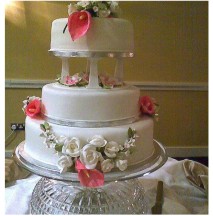

🤖 **Initial Response**

Yes, there is a large, four-tiered wedding cake in the image, and it is adorned after being done. The cake features unique details like pink flowers. It stands on top of a white cake plate, and the overall appearance is perfect for a special event, such as a wedding.

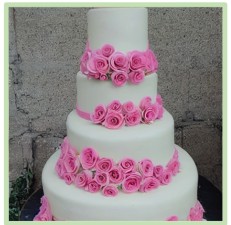

✔ Correct Generation

Query: Is there a kite in this image? Describe relevant details about this.

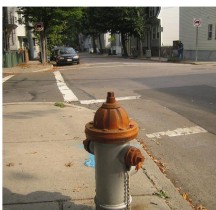

🤖 **Initial Response**

Yes, there is a kite that can be seen in the image. It is being flown by someone in an open yard near the fire hydrant. This scene serves as a contrast between the urban setting with the fire hydrant and pole, and the more recreational activity taking place in the kite flying.

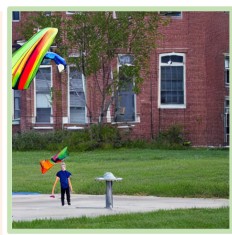

✔ Correct Generation

Query: Is there a car in this image? Describe relevant details about this.

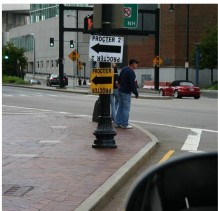

🤖 **Initial Response**

Yes, there is a car in the image. There is a red car on the street, which is parked next to some buildings and traffic lights. The car appears to be lining up to enter a business area behind a man standing near the street crossing sign.

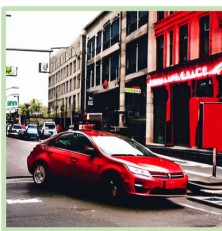

✔ Correct Generation

Query: Is there a skateboard in this image? Describe relevant details about this.

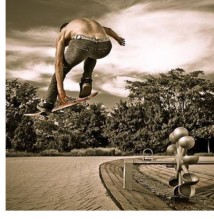

🤖 **Initial Response**

Yes, there is a skateboard in this image. A man is jumping mid-air on the skateboard with his hands and feet out. The skateboard and the man on it are in motion, which adds to the excitement and action of the scene.

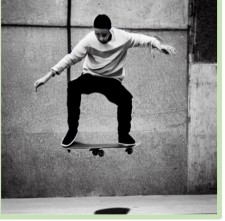

✔ Correct Generation

Query: Is there a mouse in this image? Describe relevant details about this.

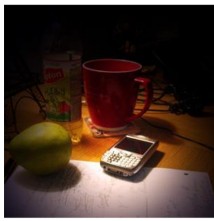

🤖 **Initial Response**

There is a mouse in the image, but it appears to be a cell phone rather than a regular computer mouse. It is placed together with the phone and other items, such as a paper, a glass, and possibly a keyboard, on the desk.

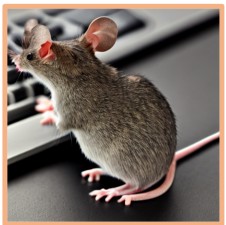

✖ Incorrect Generation

Figure D3: **Qualitative evaluation of images generated by the generative model on the *existence* subset of the MME benchmark.** Specifically, the left displays the original image input, the middle presents the initial response generated by the LVLMs, and the right shows the image generated based on this response. This figure showcases four success cases and one failure case of our Diffusion models in generating high-quality images that align with the initial response.

**Query:** Are there a total of two dogs in the picture? Describe relevant details about this.

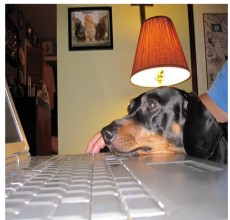

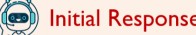 **Initial Response**

Yes, there are a total of two dogs in the picture. One of the dogs is lying down and looking at the computer, resting its head on the keyboard. The other dog is visible just behind or to the side of the main dog.

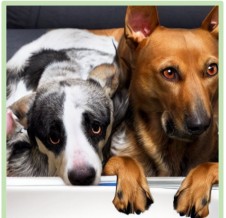

✔ Correct Generation

**Query:** Is there an elephant in this image? Describe relevant details about this.

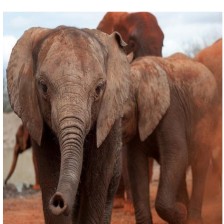

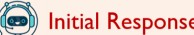 **Initial Response**

Yes, there is an elephant in the image. It is a baby elephant with a gray and muddy coat, standing in mud with its ears spread out.

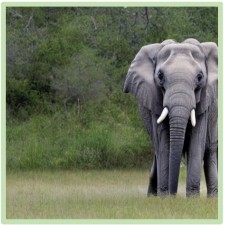

✔ Correct Generation

**Query:** Are there two bowls in this image? Describe relevant details about this.

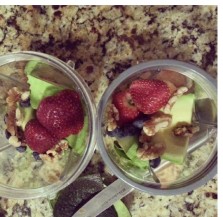

**Initial Response**

Yes, there are two bowls in the image, and both are filled with a fruit salad containing fruits such as strawberries, blueberries, mango, and walnuts. The bowls are sitting on a table or counter top.

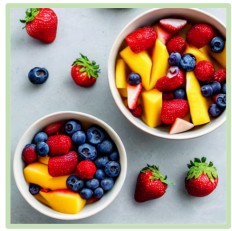

✔ Correct Generation

**Query:** Are there three remotes in this image? Describe relevant details about this.

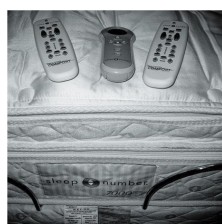

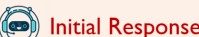 **Initial Response**

Yes, there are three remotes in this image. Two of them are white, similar to a Nintendo Wii remote, while the third one has a more traditional-looking design. They are placed on top of a bed, which has a white handle and a comforter on top.

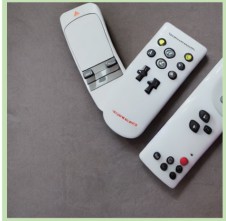

✔ Correct Generation

**Query:** Are there three neckties in the image? Describe relevant details about this.

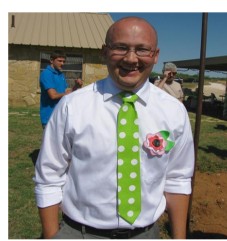

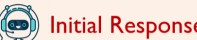 **Initial Response**

Yes, there is an image of a bald man wearing three different neckties, indicating a playful and unique sense of style.

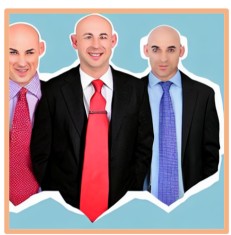

✘ Incorrect Generation

Figure D4: **Qualitative evaluation of images generated by the generative model on the *count subset of the MME benchmark.*** Specifically, the left displays the original image input, the middle presents the initial response generated by the LVLMs, and the right shows the image generated based on this response. This figure showcases four success cases and one failure case of our Diffusion models in generating high-quality images that align with the initial response.

**Query:** Are the pedestrians on the left of the bus? Describe relevant details about this.

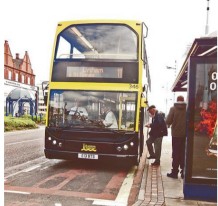

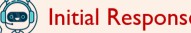 Initial Response

Yes, the pedestrians are on the left side of the bus. One pedestrian is standing near the road, while two other people are waiting in line to board the yellow city bus.

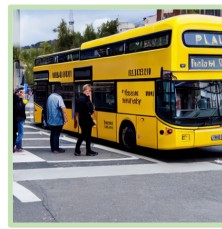

✔ Correct Generation

**Query:** Is the TV on the right of the lamp? Describe relevant details about this.

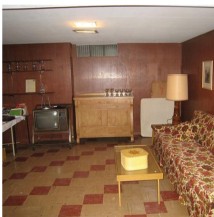

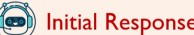 Initial Response

Yes, the TV is on the right of the lamp. The room appears to be old-style, and there is a couch set accompanied by a television. There are also several TV remotes on the table near the couch, indicating the presence of multiple control options for the TV.

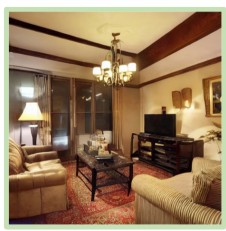

✔ Correct Generation

**Query:** Is the refrigerator on the left side of the picture? Describe relevant details about this.

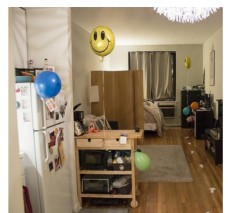

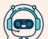 Initial Response

Yes, the refrigerator is located on the left side of the picture. It has a picture placed on the refrigerator door.

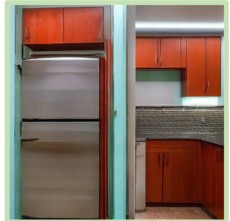

✔ Correct Generation

**Query:** Is the sea behind people in the image? Describe relevant details about this.

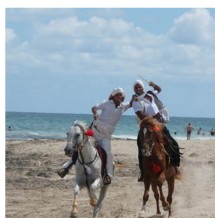

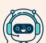 Initial Response

Yes, the sea is behind people in the image. Located in a sandy beach area, the two men are riding horses, and in the distance, there is an ocean, providing a picturesque and serene setting for the scene.

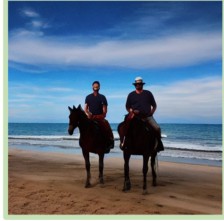

✔ Correct Generation

**Query:** Is the vase on the right of the bottle? Describe relevant details about this.

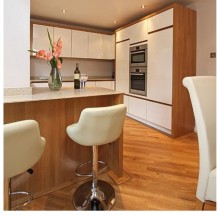

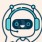 Initial Response

Yes, the vase is on the right of the bottle. It sits on a table near the kitchen counter with a tray holding the gourmet food.

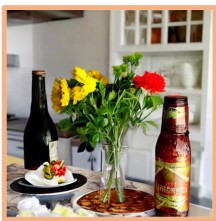

✘ Incorrect Generation

Figure D5: **Qualitative evaluation of images generated by the generative model on the *position* subset of the MME benchmark.** Specifically, the left displays the original image input, the middle presents the initial response generated by the LVLMs, and the right shows the image generated based on this response. This figure showcases four success cases and one failure case of our Diffusion models in generating high-quality images that align with the initial response.

Query: Is there a blue and orange fire hydrant in the image? Describe relevant details about this.

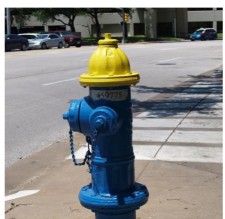

**Initial Response**

The image features a blue and orange engine company fire hydrant on the side of the street. It is a large, metal fire hydrant near an intersection.

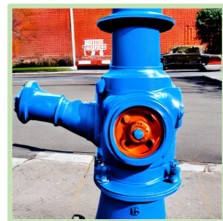

✔ Correct Generation

Query: Is there a red hat in the image? Describe relevant details about this.

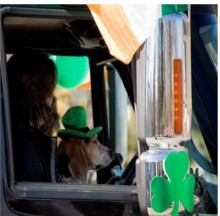

**Initial Response**

Yes, a dog wearing a red hat is a notable aspect of this image. The dog is wearing a green top hat as well, which adds to the entertaining and whimsical nature of the scene.

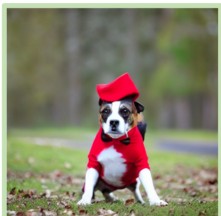

✔ Correct Generation

Query: Is there a living room painted yellow in the image? Describe relevant details about this.

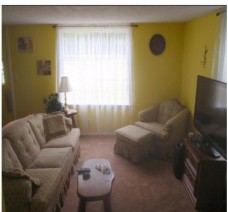

**Initial Response**

Yes, the image shows a living room that is painted yellow. It is an old yellow living room.

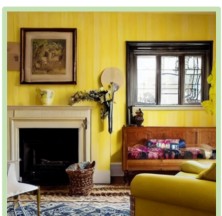

✔ Correct Generation

Query: Is there a brown and black dog in the image? Describe relevant details about this.

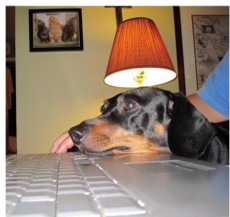

**Initial Response**

Yes, there is a brown and black dog given the name Max in the image, which is likely a Dachshund. This dog is laying on its side next to a person, possibly on a bed or a desk with a laptop.

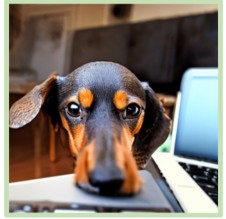

✔ Correct Generation

Query: Is there a red bicycle with white handlebars in the image? Describe relevant details about this.

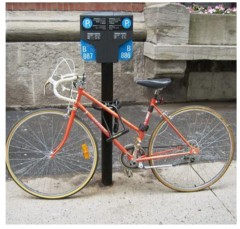

**Initial Response**

Yes, there is an old red bicycle sitting locked up against a pole, featuring pink handlebars. It is parked underneath a parking meter, indicating that the owner has tied it to the bike rail to ensure its security while they are away.

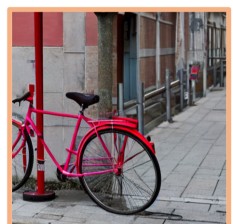

✗ Incorrect Generation

Figure D6: **Qualitative evaluation of images generated by the generative model on the *color* subset of the MME benchmark.** Specifically, the left displays the original image input, the middle presents the initial response generated by the LVLMs, and the right shows the image generated based on this response. This figure showcases four success cases and one failure case of our Diffusion models in generating high-quality images that align with the initial response.

