# OpenReview forum: "Self-Correcting Decoding with Generative Feedback for Mitigating Hallucinations in Large Vision-Language Models"
_ICLR.cc/2025/Conference — ICLR 2025 Poster_

### Official Review · Reviewer_prZa · 2024-10-24

**Soundness:** 3
**Presentation:** 3
**Contribution:** 3
**Rating:** 6
**Confidence:** 4

**Summary:**

The study proposes a new decoding method called Decoding with Generative Feedback (DeGF), which utilizes self-correcting feedback from text-to-image generative models to improve accuracy. By comparing original visual content with generated visuals based on responses, DeGF refines outputs through contrastive predictions. Extensive evaluations show that DeGF enhances performance across benchmarks, effectively reducing various types of hallucinations. However, DeGF faces limitations due to its time-consuming image generation process, limited benchmark evaluations, and inconsistencies in experimental results—particularly on the POPE benchmark compared to LLaVA-1.5—raising concerns about its suitability for real-time use and overall reliability.

**Strengths:**

- DeGF Leverages generative models to provide self-feedback by comparing the original visual input with a newly generated image, which helps in correcting hallucinations at both response and token levels.

- DeGF  demonstrates the capability of DeGF to handle diverse hallucination types, including object existence, visual appearance, and counting, showing generalizability beyond just language biases.

- DeGF  offers an efficient, cost-effective alternative to existing methods as it avoids the need for additional data or training, focusing instead on refining the decoding process.

- The study finds that text-to-image generative models effectively identify hallucinations. Higher discrepancies in token predictions, measured by JS divergence, indicate hallucinations, showing that generative feedback can highlight inaccuracies both broadly and specifically.

**Weaknesses:**

- The process of generating images adds time, potentially slowing down LVLM response generation and making it less suitable for real-time applications.
- The authors did not perform more extensive evaluations on comprehensive benchmark such as MMbench, which is crucial for assessing the method's overall performance.
- I am confused by the experimental results about POPE, as they do not seem to fully align with the result from LLAVA 1.5.

**Questions:**

-  Could the authors clarify the specific settings followed in the experiments on POPE benchmark?  How do these settings differ from those used in LLaVA?
- Could authors follow LLaVA’s setting and conduct more extensive evaluations on comprehensive benchmarks like MMbench and MMVet, given their importance for assessing the model's overall performance?
- Is this decoding method useful in more advanced VLLMs, such as Qwen-VL, Intern-VL, etc.?

---

> ### Author Response · Authors · 2024-11-21
> **Response to Reviewer prZa (Part 1/3)**
>
> Dear Reviewer prZa,
>
> We greatly appreciate your valuable feedback on our paper. We address the raised concerns and questions below.
>
> ---
>
> **Comment (1)**: “*The process of generating images adds time, potentially slowing down LVLM response generation and making it less suitable for real-time applications*.”
>
> **Response (1)**: Thanks for pointing out this. We will respond from two aspects:
>
> - **Efficiency Comparison**. Following your comments, we compare the efficiency of our approach with other state-of-the-art approaches such as OPERA, Woodpecker, and HALC. Specifically, we conduct this comparison on the CHAIR benchmark, with the maximum token length set to 128. The results are as follows:
>
>     | Method     | Avg. Latency $\downarrow$ | GPU Memory $\downarrow$  | CHAIR$_S$ $\downarrow$ |
>     | ---------- | ------------------------- | ------------------------ | ---------------------- |
>     | Regular    | 3.44 s ($\times$1.00)     | 15778 MB ($\times$1.00)  | 55.0  |
>     | VCD        | 6.91 s ($\times$2.01)     | 16634 MB ($\times$1.05) | 54.4 |
>     | OPERA      | 24.70 s ($\times$7.18)    | 22706 MB ($\times$1.44)  | 52.6  |
>     | Woodpecker | 10.68 s ($\times$3.10)     | 22199 MB ($\times$1.41)  | 57.6 |
>     | HALC       | 22.61 s ($\times$6.51)    | 23084 MB ($\times$1.46)  | 51.0 |
>     | Ours       | 13.89 s ($\times$4.04)    | 19119 MB  ($\times$1.21) | 48.8  |
>
>     Our approach involves two queries and incorporates a text-to-image generation model to mitigate hallucinations, resulting in a 4.04$\times$ increase in latency and a 1.21$\times$ increase in GPU memory usage. Compared to other approaches, while our method is slower than regular decoding and contrastive decoding-based methods, it demonstrates efficiency advantages over OPERA and HALC. Note that our approach also outperforms the compared methods in performance on the CHAIR benchmark.
>
> - **Strategies to Speed Up Our Approach**. To further enhance the efficiency of our approach, we propose two strategies to accelerate our approach: limiting the length of the initial response and reducing the number of inference steps in the diffusion process.
>
>     **(1) Reducing Diffusion Inference Steps**. In the following table, we report the performance on the CHAIR benchmark after reducing the diffusion inference steps in the model. By reducing the diffusion inference steps from 50 to 10, the average latency decreases by 2.85 seconds per instance, while the performance on CHAIR remains robust.
>
>     | Diffusion Steps | Avg. Latency $\downarrow$ | CHAIR$_S$ $\downarrow$ | CHAIR$_I$     $\downarrow$ |
>     | --------------- | ------------------------- | -------------------- | -------------------- |
>     | 50   | 13.89 s   | 48.8    | 14.6|
>     | 30   | 12.56 s   | 48.9   | 14.7   |
>     | 20 | 11.87 s   | 49.2   | 14.8    |
>     | 10    | 11.04 s    | 48.8   | 14.9    |
>
>     **(2) Restricting Length of Initial Response**. In the following table, we present the efficiency and CHAIR performance results after decreasing the maximum token limit for the initial response. We can see that reducing the maximum number of tokens in the initial response from 128 to 96 decreases the latency by 0.72 seconds per instance while maintaining competitive performance. However, further reductions result in performance degradation, as a shorter initial response fails to adequately cover the entire scene, limiting its ability to generate an image that effectively reflects and mitigates hallucinations.
>
>     | # Tokens | Avg. Latency $\downarrow$ | CHAIR$_S$ $\downarrow$ | CHAIR$_I$ $\downarrow$ |
>     | -------- | ------------------------- | -------------------- | -------------------- |
>     | 128 | 13.89 s  | 48.8  | 14.6   |
>     | 96  | 13.17 s  | 48.8 | 14.9  |
>     | 64   | 12.20 s | 49.5 | 14.8 |
>     | 32  | 11.33 s | 51.2  | 14.9  |
>
>     Note that these two strategies are not conflicting; instead, they are complementary. Setting the diffusion steps to 10 and limiting the maximum number of tokens in the initial response to 96 further reduces the inference latency to 10.21 seconds per instance while maintaining robust performance.
>
> We have also added these discussions and additional experiments to Section 4.4 and Appendix C.8 of the revised paper. Again, thank you for your valuable feedback to improve our paper.

---

> ### Author Response · Authors · 2024-11-21
> **Response to Reviewer prZa (Part 2/3)**
>
> **Comment (2)**: “*The authors did not perform more extensive evaluations on comprehensive benchmark such as MMbench, which is crucial for assessing the method's overall performance*.”
>
> **Response (2)**: Following your comments, we have conducted an additional experiment to assess our approach on the MMBench benchmark. The overall performance results are presented below:
>
> | Method  | Regular | DoLa | OPERA | VCD  | M3ID | RITUAL | Ours |
> | ------- | ------- | ---- | ----- | ---- | ---- | ------ | ---- |
> | MMBench | 64.1    | 63.8 | 64.4  | 64.6 | 64.4 | 63.8   | **65.5** |
>
> As shown in the table, our approach achieves superior performance compared to the baseline methods on the MMBench benchmark, highlighting the effectiveness of our proposed method. We have incorporated these results into Table 3 of the revised paper.
>
> In the following table, we also present the detailed performance across six L-2 abilities on the MMBench benchmark: Logical Reasoning (LR), Attribute Reasoning (AR), Relation Reasoning (RR), Fine-grained Perception - Single Instance (FP-S), Fine-grained Perception - Cross Instance (FP-C), and Coarse Perception (CP). We have also included these results in Appendix C.3.
>
> | Method | LR | AR | RR | FP-S | FP-C | CP | Overall |
> |----------|--------|--------|--------|--------|--------|--------|---------|
> | Regular | 30.51 | 71.36 | 52.17 | 67.58 | **58.74** | 76.35 | 64.09 |
> | VCD | 30.51 | **73.37** | 53.04 | 67.92 | 57.34 | 77.03 | 64.60 |
> | M3ID | 30.51 | 72.36 | 53.04 | 67.58 | 57.34 | **77.36** | 64.43 |
> | RITUAL | 28.81 | 72.86 | 54.78 | 65.87 | 58.04 | 76.01 | 63.83 |
> | **Ours** | **31.36** | 70.85 | **60.87** | **68.60** | **58.74** | **77.36** | **65.46** |
>
>
> ---
>
> **Comment (3)**: “*I am confused by the experimental results about POPE, as they do not seem to fully align with the result from LLAVA 1.5. Could the authors clarify the specific settings followed in the experiments on POPE benchmark? How do these settings differ from those used in LLaVA?*”
>
> **Response (3)**: In the LLaVA-1.5 paper [1], the model is evaluated using greedy decoding, as stated in their Appendix A.3, where each generated token is selected based on the maximum prediction probability. In contrast, our experiments evaluate all approaches using regular decoding, where the final generated token $y_t$ is sampled from the multinomial distribution with probabilities $p_\theta(y_t)$.This difference leads to performance discrepancies between our evaluation and that in the LLaVA-1.5 paper. Note that regular decoding is a standard benchmarking setting, and we follow prior work [2, 3] in using this setting for evaluation. We have clarified this setting in the Baselines section on page 6 of the revised paper.
>
> Following your comments, we have also conducted an additional experiments using greedy decoding on the POPE benchmark on the COCO random subset:
>
> | Method   | Acc.      | Prec.     | Rec.      | F1        |
> | -------- | --------- | --------- | --------- | --------- |
> | Greedy   | 87.73     | 88.19     | 87.13     | 87.66     |
> | VCD      | 87.47     | 86.64     | **88.60** | 87.61     |
> | M3ID     | 89.07     | 89.54     | 88.47     | 89.00     |
> | RITUAL   | 89.23     | 90.17     | 88.07     | **89.11** |
> | **Ours** | **89.40** | **94.44** | 83.73     | 88.76     |
>
> We can see that using greedy decoding improves the performance of all baseline methods as well as our proposed DeGF, and our approach consistently achieves state-of-the-art performance. We have also added this experiment to Appendix C.4.
>
> [1] Liu, H., Li, C., Li, Y., & Lee, Y. J. (2024). Improved baselines with visual instruction tuning. In *Proceedings of the IEEE/CVF Conference on Computer Vision and Pattern Recognition* (pp. 26296-26306).
>
> [2] Leng, S., Zhang, H., Chen, G., Li, X., Lu, S., Miao, C., & Bing, L. (2024). Mitigating object hallucinations in large vision-language models through visual contrastive decoding. In *Proceedings of the IEEE/CVF Conference on Computer Vision and Pattern Recognition* (pp. 13872-13882).
>
> [3] Woo, S., Jang, J., Kim, D., Choi, Y., & Kim, C. (2024). *RITUAL: Random Image Transformations as a Universal Anti-hallucination Lever in LVLMs*. arXiv preprint arXiv:2405.17821.

---

> ### Author Response · Authors · 2024-11-21
> **Response to Reviewer prZa (Part 3/3)**
>
> **Comment (4)**: “*Is this decoding method useful in more advanced VLLMs, such as Qwen-VL, Intern-VL, etc.?*”
>
> **Response (4)**: Thank you for your thoughtful comments. Indeed, our approach can be applied to more advanced LVLMs and effectively mitigates hallucinations in these models. Here, we use Qwen-VL as an example and evaluate its performance on the POPE benchmark:
>
> ### POPE Benchmark with Qwen-VL Backbone:
> **Random Setup**
> | Method    | Acc. $\uparrow$ | Prec. $\uparrow$ | Rec. $\uparrow$ | F1 $\uparrow$  |
> |-----------|---------|----------|--------|-------|
> | Regular   | 87.43   | 93.56    | 80.40   | 86.48 |
> | VCD       | 88.80    | 93.89    | 83.00   | 88.11 |
> | M3ID      | 89.83   | 95.44    | 83.65  | 89.17 |
> | RITUAL    | 89.47   | **96.32**    | 82.07  | 88.62 |
> | **Ours**  | **89.73** | 93.19  | **85.73** | **89.31** |
>
> **Popular Setup**
> | Method    | Acc. $\uparrow$ | Prec. $\uparrow$ | Rec. $\uparrow$ | F1 $\uparrow$  |
> |-----------|---------|----------|--------|-------|
> | Regular   | 84.70    | 88.24    | 80.07  | 83.96 |
> | VCD       | 85.13   | 87.27    | 82.27  | 84.69 |
> | M3ID      | 86.27   | 89.19    | 82.53  | **85.73** |
> | RITUAL    | 84.57   | 84.09    | **85.27** | 84.67 |
> | **Ours**  | **86.5** | **89.87** | 81.93  | 85.71 |
>
> **Adversarial Setup**
> | Method    | Acc. $\uparrow$ | Prec. $\uparrow$ | Rec. $\uparrow$ | F1 $\uparrow$  |
> |-----------|---------|----------|--------|-------|
> | Regular   | 79.83   | 80.13    | 79.33  | 79.73 |
> | VCD       | 81.33   | 80.60     | 82.53  | 81.55 |
> | M3ID      | 82.03   | 81.47    | **82.93** | 82.19 |
> | RITUAL    | 82.80    | 83.15    | 82.27  | 82.71 |
> | **Ours**  | **83.47** | **84.49** | 81.52  | **82.98** |
>
> We have provided the full results in Table 1 of the revised paper. Please kindly refer to it for the results on A-OKVQA and GQA datasets.
>
> On the MME-Hallucination benchmark, our approach also achieves state-of-the-art results on the Qwen-VL benchmark, as presented below. We have also incorporated this into Table C2 of the Appendix (due to the page limit).
>
> | Method   | Existence $\uparrow$      | Count $\uparrow$         | Position $\uparrow$       | Color $\uparrow$         | Total Score $\uparrow$    |
> |----------|---------------------------|--------------------------|---------------------------|--------------------------|---------------------------|
> | Regular  | 155.00 (±3.54)           | 127.67 (±13.36)         | 131.67 (±7.73)           | 173.00 (±9.75)          | 587.33 (±31.06)          |
> | VCD      | 156.00 (±6.52)           | 131.00 (±6.19)          | 128.00 (±3.61)           | 181.67 (±5.14)          | 596.67 (±11.61)          |
> | M3ID     | 178.33 (±2.89)           | 143.33 (±2.89)          | 150.00 (±2.89)           | 175.00 (±5.00)          | 646.66 (±8.50)           |
> | RITUAL   | 178.33 (±2.89)           | 142.22 (±16.19)         | **156.66** (±2.89)           | **178.33** (±2.89)          | 655.55 (±14.99)          |
> | Ours     | **180.00** (±0.00)           | **148.89** (±6.74)          | 155.00 (±7.64)           | **178.33** (±2.89)          | **662.22** (±4.37)           |
>
>
> In summary, these results demonstrate the general effectiveness of our approach across various LVLMs.
>
> ---
>
> We hope that our responses have addressed your concerns. If you have additional comments or concerns, please let us know and we will be more than happy to answer.
>
>
> Best,
>
> Authors

---

> ### Author Response · Authors · 2024-11-25
>
> Dear Reviewer prZa,
>
> We greatly appreciate the time you have dedicated and the valuable feedback you have provided. As the discussion period draws to a close (Tue, November 26), please kindly let us know if there are any remaining questions. We will be more than happy to provide any further details or clarifications.
>
> Best,
>
> Authors

---

> > ### Comment · Reviewer_prZa · 2024-11-26
> >
> > Thank you for the authors' detailed response. I decide to keep my score.

---

> > > ### Author Response · Authors · 2024-11-26
> > > **Thanks for your positive recommendation**
> > >
> > > Dear Reviewer prZa,
> > >
> > > Thank you for your thoughtful feedback. We really appreciate your positive recommendation!
> > >
> > > Best,
> > >
> > > Authors

---

> > > ### Author Response · Authors · 2024-11-27
> > > **Further Experiments on the MM-Vet Benchmark**
> > >
> > > Dear Reviewer prZa,
> > >
> > > With the extended discussion period, we had the opportunity to conduct additional experiments to address your concerns more comprehensively. Specifically, we carried out extensive evaluations on the MM-Vet benchmark using both regular sampling and greedy decoding approaches, as per your suggestion. The results are as follows:
> > >
> > >
> > > **Regular Sampling on MMVet Benchmark**
> > >
> > > | Method  | Rec      | OCR      | Know     | Gen      | Spat     | Math     | Total    |
> > > | ------- | -------- | -------- | -------- | -------- | -------- | -------- | -------- |
> > > | Regular | 30.8     | 19.0     | 14.5     | 17.9     | 26.9     | **11.5** | 26.1     |
> > > | VCD     | 35.6     | 21.9     | 18.3     | 21.9     | 28.9     | 3.8      | 30.9     |
> > > | M3ID    | 35.0     | 19.7     | 18.8     | 19.0     | 26.0     | 7.7      | 29.9     |
> > > | RITUAL  | **36.3** | 20.6     | **19.5** | 21.1     | 24.7     | 7.7      | 30.6     |
> > > | Ours    | 35.9     | **27.2** | 19.2     | **22.4** | **30.4** | **11.5** | **33.0** |
> > >
> > > **Greedy Decoding on MMVet Benchmark**
> > >
> > > | Method | Rec | OCR | Know | Gen | Spat | Math | Total |
> > > |--------|-------|-------|-------|-------|-------|-------|-------|
> > > | Greedy | 37.0 | 22.6 | 17.5 | 20.2 | 24.9 | 7.7 | 31.8 |
> > > | VCD | **38.2** | 22.8 | 22.5 | **24.6** | 25.1 | 3.8 | 33.4 |
> > > | M3ID | 37.9 | 23.6 | **20.4** | 20.7 | 26.0 | 11.5 | 33.2 |
> > > | RITUAL | 35.6 | 21.7 | 18.9 | 19.9 | 24.7 | 7.7 | 30.6 |
> > > | Ours | 37.9 | **25.0** | 20.2 | 19.5 | **32.8** | **15.0** | **34.0** |
> > >
> > > From the results, we observed that our method consistently outperforms others on the MMVet benchmark. Notably, it significantly excels in the OCR, spatial awareness, and math subsets. We have incorporated these results into Table C5 and Table C6 of the revised paper.
> > >
> > > We sincerely appreciate your thoughtful comments, which have helped us improve the paper. If you have any further questions or concerns, please don’t hesitate to let us know. We would also be grateful if you could consider raising your score based on these additional improvements.
> > >
> > > Best,
> > >
> > > Authors

---

### Official Review · Reviewer_QadT · 2024-10-27

**Soundness:** 3
**Presentation:** 3
**Contribution:** 3
**Rating:** 6
**Confidence:** 4

**Summary:**

This paper proposes self-correcting decoding with generative feedback (DeGF), a novel training-free approach that leverages feedback from diffusion models and corrects decoding procedure, either by contrastive or complementary manner. This paper has well grounded their approach on thorough analysis and ablations, surpassing previous approaches on five hallucination benchmarks.

**Strengths:**

Clear writing. Presentation and writing are clear and easy to follow.

Well-motivated. A clear negative correlation between hallucination rates and CLIP similarities can be observed (Figure 3), which gives a strong empirical foundation of proposed decoding approach.

Sufficient experiments. Authors do sufficient ablations, discussions to support their claims. Performance of proposed DeGF is quite good.

**Weaknesses:**

A concern originated from numerical hallucinations. Figure 2 presents an overview of proposed approach DeGF, with addressing numeric hallucinations as an example. A key premise for this method is that diffusion models can accurately perceive numbers. However, it seems a common observation that diffusion models fail to accurately interpret numbers [A]. Have authors considered the case when diffusion models fail to generate numerically accurate images? It is good to include analysis on tolerance of proposed DeGF to diffusion model failures.

Increased test-time computation cost. Diffusion models take dozens of inference steps to generate images, which is computationally less efficient than other decoding approaches.

Authors are suggested to include test-time computation comparison to complete this work.

[A] Mass-Producing Failures of Multimodal Systems with Language Models. NeurIPS 2023.

**Questions:**

1. May I ask how many samples authors use for plotting Figure 3?

2. For Table 2, may I ask if authors have tested long sequence, by setting max new tokens to larger numbers, such as 128 and 256?

---

> ### Author Response · Authors · 2024-11-21
> **Response to Reviewer QadT (Part 1/3)**
>
> Dear Reviewer QadT,
>
> Thank you for your insightful comments and positive recommendation of our work. We provide point-by-point responses to address your concerns below.
>
> ---
>
> **Comment (1)**: “*A concern originated from numerical hallucinations. Figure 2 presents an overview of proposed approach DeGF, with addressing numeric hallucinations as an example. A key premise for this method is that diffusion models can accurately perceive numbers. However, it seems a common observation that diffusion models fail to accurately interpret numbers [A]. Have authors considered the case when diffusion models fail to generate numerically accurate images? It is good to include analysis on tolerance of proposed DeGF to diffusion model failures*.”
>
> **Response (1)**: Thank you for your thoughtful feedback. Following your comments, we evaluated the quality of the generated images across all four subsets of the MME benchmark, including the "count" subset, which focuses on numerical hallucinations, using CLIPScore [1]. For this evaluation, we employed the CLIP model with the ViT-B/32 backbone. The results are summarized below:
>
> | MME Subset     | Existence | Count | Position | Color |
> | -------------- | --------- | ----- | -------- | ----- |
> | Avg. CLIPScore | 31.34     | 30.69 | 30.09    | 31.69 |
>
> As we can see from the table, our text-to-image generative model (specifically, SD-v1.5) achieves an average CLIPScore of over 30 across all subsets, demonstrating its ability to generate high-quality images that align well with the initial response. For comparison, the advanced DALL-E 3 model achieves a score of 32.0, while DALL-E 2 achieves 31.4 [2]. While, as you pointed out, our diffusion model performs worse on the count subset compared to the other two subsets, a CLIPScore of 30.69 indicates that the generated images are of sufficient quality to provide useful self-feedback. We have included this discussion in Appendix C.9.
>
> Additionally, in Figure D3-D6, we provide qualitative evaluations of the images generated by the generative model, including both success and failure cases, across all four subsets of the MME benchmark to better understand the effectiveness of the generative models. Our results in Figure D4 show that, despite occasional failure cases, the generative model can accurately interpret numbers and produce high-quality and realistic images that accurately visualize the initial response. Moreover, our substantial performance improvement (e.g., ~30 points higher than regular decoding) on the count subset of the MME benchmark, as presented in Table 3, demonstrates the robustness of our approach to potential image generation failures and validates the effectiveness of our method.
>
>
> [1] Hessel, J., Holtzman, A., Forbes, M., Le Bras, R., & Choi, Y. (2021, November). CLIPScore: A Reference-free Evaluation Metric for Image Captioning. In *Proceedings of the 2021 Conference on Empirical Methods in Natural Language Processing* (pp. 7514-7528).
>
> [2] Betker, J., Goh, G., Jing, L., Brooks, T., Wang, J., Li, L., ... & Ramesh, A. (2023). Improving image generation with better captions. https://cdn.openai.com/papers/dall-e-3.pdf

---

> ### Author Response · Authors · 2024-11-21
> **Response to Reviewer QadT (Part 2/3)**
>
> **Comment (2)**: “*Increased test-time computation cost. Diffusion models take dozens of inference steps to generate images, which is computationally less efficient than other decoding approaches. Authors are suggested to include test-time computation comparison to complete this work.*”
>
> **Response (2)**:  Thanks for pointing out this. We will respond from two aspects:
>
> - **Efficiency Comparison**. Following your comments, we compare the efficiency of our approach with other state-of-the-art approaches such as OPERA, Woodpecker, and HALC. Specifically, we conduct this comparison on the CHAIR benchmark, with the maximum token length set to 128. The results are as follows:
>
>     | Method     | Avg. Latency $\downarrow$ | GPU Memory $\downarrow$  | CHAIR$_S$ $\downarrow$ |
>     | ---------- | ------------------------- | ------------------------ | ---------------------- |
>     | Regular    | 3.44 s ($\times$1.00)     | 15778 MB ($\times$1.00)  | 55.0  |
>     | VCD        | 6.91 s ($\times$2.01)     | 16634 MB ($\times$1.05) | 54.4 |
>     | OPERA      | 24.70 s ($\times$7.18)    | 22706 MB ($\times$1.44)  | 52.6  |
>     | Woodpecker | 10.68 s ($\times$3.10)     | 22199 MB ($\times$1.41)  | 57.6 |
>     | HALC       | 22.61 s ($\times$6.51)    | 23084 MB ($\times$1.46)  | 51.0 |
>     | Ours       | 13.89 s ($\times$4.04)    | 19119 MB  ($\times$1.21) | 48.8  |
>
>     Our approach involves two queries and incorporates a text-to-image generation model to mitigate hallucinations, resulting in a 4.04$\times$ increase in latency and a 1.21$\times$ increase in GPU memory usage. Compared to other approaches, while our method is slower than regular decoding and contrastive decoding-based methods, it demonstrates efficiency advantages over OPERA and HALC. Note that our approach also outperforms the compared methods in performance on the CHAIR benchmark.
>
> - **Strategies to Speed Up Our Approach**. To further enhance the efficiency of our approach, we propose two strategies to accelerate our approach: limiting the length of the initial response and reducing the number of inference steps in the diffusion process.
>
>     **(1) Reducing Diffusion Inference Steps**. In the following table, we report the performance on the CHAIR benchmark after reducing the diffusion inference steps in the model. By reducing the diffusion inference steps from 50 to 10, the average latency decreases by 2.85 seconds per instance, while the performance on CHAIR remains robust.
>
>     | Diffusion Steps | Avg. Latency $\downarrow$ | CHAIR$_S$ $\downarrow$ | CHAIR$_I$     $\downarrow$ |
>     | --------------- | ------------------------- | -------------------- | -------------------- |
>     | 50   | 13.89 s   | 48.8    | 14.6|
>     | 30   | 12.56 s   | 48.9   | 14.7   |
>     | 20 | 11.87 s   | 49.2   | 14.8    |
>     | 10    | 11.04 s    | 48.8   | 14.9    |
>
>     **(2) Restricting Length of Initial Response**. In the following table, we present the efficiency and CHAIR performance results after decreasing the maximum token limit for the initial response. We can see that reducing the maximum number of tokens in the initial response from 128 to 96 decreases the latency by 0.72 seconds per instance while maintaining competitive performance. However, further reductions result in performance degradation, as a shorter initial response fails to adequately cover the entire scene, limiting its ability to generate an image that effectively reflects and mitigates hallucinations.
>
>     | # Tokens | Avg. Latency $\downarrow$ | CHAIR$_S$ $\downarrow$ | CHAIR$_I$ $\downarrow$ |
>     | -------- | ------------------------- | -------------------- | -------------------- |
>     | 128 | 13.89 s  | 48.8  | 14.6   |
>     | 96  | 13.17 s  | 48.8 | 14.9  |
>     | 64   | 12.20 s | 49.5 | 14.8 |
>     | 32  | 11.33 s | 51.2  | 14.9  |
>
>     Note that these two strategies are not conflicting; instead, they are complementary. Setting the diffusion steps to 10 and limiting the maximum number of tokens in the initial response to 96 further reduces the inference latency to 10.21 seconds per instance while maintaining robust performance.
>
> We have also added these discussions and additional experiments to Section 4.4 and Appendix C.8 of the revised paper. Again, thank you for your valuable feedback to improve our paper.

---

> ### Author Response · Authors · 2024-11-21
> **Response to Reviewer QadT (Part 3/3)**
>
> **Comment (3)**: “*May I ask how many samples authors use for plotting Figure 3?*”
>
>
> **Response (3)**: For Figure 3 (*Left*), we evaluate using 1,000 samples from the CHAIR benchmark, while for Figure 3 (*Right*), we use all 3,000 samples from the POPE benchmark using the random subset from the MS-COCO dataset. We have added a footnote at the end of Page 4 to clarify this.
>
> This experiment validates the effectiveness of text-to-image generative models in reflecting hallucinations. It demonstrates that these models can provide valuable self-feedback for mitigating hallucinations at both the response and token levels.
>
> ---
>
>
> **Comment (4)**: “*For Table 2, may I ask if authors have tested long sequence, by setting max new tokens to larger numbers, such as 128 and 256?*”
>
> **Response (4)**: Following your comments, we have conducted an additional experiment by setting the maximum token limit to 128 in the CHAIR benchmark. The results are reported below:
>
> ### Results with LLaVA-1.5 backbone:
>
> | Method  | CHAIR$_S$ $\downarrow$ | CHAIR$_I$ $\downarrow$ | Recall $\uparrow$ | Length $\uparrow$ |
> | ------- | ---------------------- | ---------------------- | ----------------- | ----------------- |
> | Regular | 55.0                   | 16.3                   | 71.9              | **97.3**          |
> | VCD     | 54.4                   | 16.6                   | 75.1              | 97.0              |
> | M3ID    | 56.6                   | 15.7                   | **79.8**          | 94.5              |
> | RITUAL  | 49.6                   | 14.8                   | 74.7              | 96.2              |
> | Ours    | **48.8**               | **14.6**               | 76.0              | 96.4              |
>
> ### Results with InstructBLIP backbone:
> | Method  | CHAIR$_S$ $\downarrow$ | CHAIR$_I$ $\downarrow$ | Recall $\uparrow$ | Length $\uparrow$ |
> | ------- | ---------------------- | ---------------------- | ----------------- | ----------------- |
> | Regular | 57.0                   | 17.6                   | 68.3              | **100.4**         |
> | VCD     | 60.4                   | 17.8                   | **72.5**          | 99.9              |
> | M3ID    | 62.2                   | 18.1                   | 71.9              | 99.8              |
> | RITUAL  | **48.4**               | 14.5                   | 72.2              | 100.0             |
> | Ours    | 49.2                   | **14.4**               | 72.2              | 98.9              |
>
> The results indicate that our approach also achieves competitive performance across two LVLMs in mitigating hallucinations during long-sequence generation scenarios. We have incorporated these additional results into Table C1 of the revised paper.
>
> ---
>
> We hope that our responses have addressed your concerns. If you have additional comments or concerns, please let us know and we will be more than happy to answer.
>
> Best,
>
> Authors

---

> > ### Comment · Reviewer_QadT · 2024-11-22
> >
> > Thanks for the authors' detailed response and clear explanation. I will keep my positive rate.

---

> > > ### Author Response · Authors · 2024-11-23
> > > **Thanks for your positive recommendation**
> > >
> > > Dear Reviewer QadT,
> > >
> > > Thank you for your encouraging positive recommendation and insightful feedback on our paper!
> > >
> > > Best,
> > >
> > > Authors

---

> > > ### Author Response · Authors · 2024-11-27
> > > **Further Experiments on the CHAIR Benchmark with a Maximum Token Limit of 256**
> > >
> > > Dear Reviewer QadT,
> > >
> > > With the extension to the discussion period, we had the opportunity to conduct additional experiments to address your concerns more thoroughly. Specifically, we conducted an additional experiment on the CHAIR benchmark, as per your suggestion, by setting the maximum number of tokens to 256 to evaluate our method’s generalizability to longer sequences:
> > >
> > > |    | **LLaVA-1.5**    |        |     |     | **InstructBLIP**   |      |     |     |
> > > |--------------|---------------------|-------------------|----------|---------|---------------------|-------------------|----------|---------|
> > > | **Method** | CHAIR$_S$ ↓          | CHAIR$_I$ ↓         | Recall ↑ | Length ↑ | CHAIR$_S$ ↓          | CHAIR$_I$ ↓         | Recall ↑ | Length ↑ |
> > > | **Regular**  | 58.0               | 17.7              | 74.1     | **106.3** | 61.0               | 18.2              | 68.9     | **112.0** |
> > > | **VCD**      | 58.2               | 16.7              | _78.0_   | _103.5_  | 63.0               | 18.6              | **72.9** | _106.3_  |
> > > | **M3ID**     | 56.8               | 16.1              | **80.7** | 98.2     | 65.8               | 19.9              | _72.4_   | 102.7    |
> > > | **RITUAL**   | _51.0_             | _15.1_            | 76.0     | 100.9    | _50.4_             | _15.3_            | 72.0     | 102.0    |
> > > | **Ours**     | **49.8**           | **14.7**          | 77.2     | 103.3    | **49.8**           | **15.1**          | 72.3     | 103.3    |
> > >
> > > Our approach continues to demonstrate consistently superior performance across two LVLMs. These results have been included in Table C2 of the revised paper.
> > >
> > > We sincerely appreciate your thoughtful comments, which have helped us improve the paper. If you have any further questions or concerns, please don’t hesitate to let us know. We would also be grateful if you could consider raising your score based on these additional improvements.
> > >
> > > Best,
> > >
> > > Authors

---

### Official Review · Reviewer_aaN5 · 2024-10-28

**Soundness:** 3
**Presentation:** 3
**Contribution:** 2
**Rating:** 6
**Confidence:** 3

**Summary:**

Recent large vision-language models (LVLMs) have demonstrated impressive performance in multi-modal tasks, but they are prone to generating hallucinated text responses that do not align with the given visual input. This work explores the potential of leveraging powerful text-to-image generative models to assist in mitigating these hallucinations. The authors discover that generative models can offer valuable self-feedback for correcting hallucinations at both the response and token levels. Building on this insight, they introduce a novel training-free decoding algorithm called "self-correcting Decoding with Generative Feedback (DeGF)" that incorporates feedback from text-to-image generative models to recursively enhance the accuracy of LVLM responses. Extensive experiments validate the effectiveness of DeGF in reducing diverse types of hallucinations, consistently outperforming state-of-the-art methods across multiple benchmarks.

**Strengths:**

1. Providing the generative feedback to mitigate hallucinations is straightforward and reasonable. Token-level refinement based on Jensen-Shannon divergence correctly utilize the generative feedback.
2. The proposed Self-Correcting Decoding with Generative Feedback (DeGF) also achieves pleasant results on POPE, CHAIR, MME, etc.
3. This paper is well-written and has clear figures. The experiments are extensive to some extent, and clearly organized.

**Weaknesses:**

1. The computation costs are unafforable for the LLM decoding strategy. Utilizing generative model like Stable Diffusion to provide generative feedback is unrealistic for practical deployment. Moreover, self-correcting decoding also consumes twice inference costs similar to contrastive decoding.
2. This approach utilizes extra pretrained network (i.e., Stable Diffusion). Baselines should contain methods that also employ extra analysis network like woodpecker [r1, r2], etc. Otherwise, it is unfair for other decoding methods.
[r] Woodpecker:Hallucinationcorrectionformultimodal largelanguage models. arXiv:2310.16045,2023.
[r2] HALC:Object hallucinationreductionviaadaptivefocal-contrastdecoding. In International Conferenceon MachineLearning, 2024.
3. No inference costs are listed and compared, which is important for LLM decoding strategies.
4. Experiments only comducted two LVLM backbones. More results like on QwenVL are better performed.

**Questions:**

1. How about using the middle step of Stable Diffusion to provide generative feedback and reduce the computation costs?
2. How about integrating the generative feedback in the instruction tuning phase? It will avoid inference costs.
3. Why not conduct GPT4/GPT4-V analysis like VCD [r1]?
[r1] Mitigating object hallucinations in large vision-language models through visual contrastive decoding. CVPR2024.

---

> ### Author Response · Authors · 2024-11-21
> **Response to Reviewer aaN5 (Part 1/3)**
>
> Dear Reviewer aaN5,
>
> Thanks for your valuable feedback! We provide point-by-point responses to address your concerns below.
>
> ---
>
> **Comment (1)**: “*The computation costs are unaffordable for the LLM decoding strategy…*”
>
> **Response (1)**: Thanks for pointing out this. Following your comments, we compare the efficiency of our approach with other state-of-the-art approaches such as OPERA, Woodpecker, and HALC. Specifically, we conduct this comparison on the CHAIR benchmark, with the maximum token length set to 128. The results are as follows:
>
> | Method     | Avg. Latency $\downarrow$ | GPU Memory $\downarrow$  | CHAIR$_S$ $\downarrow$ |
> | ---------- | ------------------------- | ------------------------ | ---------------------- |
> | Regular    | 3.44 s ($\times$1.00)     | 15778 MB ($\times$1.00)  | 55.0  |
> | VCD        | 6.91 s ($\times$2.01)     | 16634 MB ($\times$1.05) | 54.4 |
> | OPERA      | 24.70 s ($\times$7.18)    | 22706 MB ($\times$1.44)  | 52.6  |
> | Woodpecker | 10.68 s ($\times$3.10)    | 22199 MB ($\times$1.41)  | 57.6 |
> | HALC       | 22.61 s ($\times$6.51)    | 23084 MB ($\times$1.46)  | 51.0 |
> | Ours       | 13.89 s ($\times$4.04)    | 19119 MB  ($\times$1.21) | 48.8  |
>
> Our approach involves two queries and incorporates a text-to-image generation model to mitigate hallucinations, resulting in a 4.04$\times$ increase in latency and a 1.21$\times$ increase in GPU memory usage. Compared to other approaches, while our method is slower than regular decoding and contrastive decoding-based methods, it demonstrates efficiency advantages over OPERA and HALC.  We have also added these discussions and additional experiments to Section 4.4 of the revised paper.
>
> While computational complexity is important, we emphasize that performance is equally crucial to consider. Our model strikes a good balance between the two: although it is 2$\times$ slower than VCD, it achieves a significantly lower CHAIR$_S$ score by 5.8. Notably, our approach also outperforms OPERA and HALC on the CHAIR benchmark, while maintaining a lower computational complexity.
>
> ---
>
> **Comment (2)**: “*This approach utilizes extra pretrained network (i.e., Stable Diffusion). Baselines should contain methods that also employ extra analysis network like woodpecker [r1, r2], etc. Otherwise, it is unfair for other decoding methods*.”
>
> **Response (2)**: Thanks for pointing out this. Following your comments, we conducted performance comparisons on the CHAIR benchmark, evaluating our methods against other baselines, including Woodpecker and HALC (reproduced).
>
> ### CHAIR Benchmark with Maximum Number of Tokens 64
> | **Method**     | **CHAIR$_S$ ↓** | **CHAIR$_I$ ↓** | **Recall ↑** | **Length ↑** | **CHAIR$_S$ ↓**  | **CHAIR$_I$ ↓** | **Recall ↑** | **Length ↑** |
> | -------------- | --------------- | --------------- | ------------ | ------------ | ---------------- | --------------- | ------------ | ------------ |
> |    | LLaVA-1.5   |   |     |  | InstructBLIP |     |  |    |
> | Regular    | 26.2  | 9.4  | 58.5 | 53.4  | 31.2  | 11.1  | 59.0  | 53.6   |
> | VCD       | 24.4   | 7.9   | 63.3   | 54.2  | 30.0   | 10.1   | 61.8    | 54.2  |
> | M3ID       | 21.4  | 6.3 | **64.4** | 53.5   | 30.8     | 10.4   | 62.6  | 53.4   |
> | RITUAL     | 22.4  | 6.9  | 63.0  | **54.9**   | 26.6    | 8.9   | 63.4  | 55.3 |
> | Woodpecker | 24.9  | 7.5  | 60.8  | 49.7  | 31.2    | 10.8    | 62.3 | 51.3  |
> | HALC       | 21.7  | 7.1  | 63.4  | 53.4  | 24.5  | 8.0  | 63.8  | 55.1   |
> | Ours      | **18.4** | **6.1**  | 62.7  | 54.1   | **24.0**  | **7.7**  | **67.2** | **55.5**   |
>
> ### CHAIR Benchmark with Maximum Number of Tokens 128
> | **Method**     | **CHAIR$_S$ ↓** | **CHAIR$_I$ ↓** | **Recall ↑** | **Length ↑** | **CHAIR$_S$ ↓**  | **CHAIR$_I$ ↓** | **Recall ↑** | **Length ↑** |
> | -------------- | --------------- | --------------- | ------------ | ------------ | ---------------- | --------------- | ------------ | ------------ |
> |  | LLaVA-1.5   |   |   |  | InstructBLIP |    |  |   |
> | Regular    | 55.0   | 16.3  | 71.9  | **97.3**   | 57.0    | 17.6    | 68.3   | **100.4**    |
> | VCD | 54.4  | 16.6   | 75.1    | 97.0  | 60.4   | 17.8    | **72.5**   | 99.9   |
> | M3ID   | 56.6 | 15.7 | **76.8**     | 94.5   | 62.2   | 18.1| 71.9   | 99.8  |
> | RITUAL  | 49.6 | 14.8   | 74.7 | 96.2  | **48.4**   | 14.5   | 72.2  | 100.0  |
> | Woodpecker | 57.6 | 16.7  | 70.3   | 93.2  | 60.8  | 17.6  | 69.7   | 97.6 |
> | HALC | 51.0    | 14.8   | 75.3  | 95.8 | 53.8  | 15.7  | 71.9   | 99.1  |
> | Ours  | **48.8**   | **14.6**   | 76.0   | 96.4  | 49.2  | **14.4**  | 72.2       | 98.9   |
>
> We can see that our approach achieves superior performance compared to Woodpecker and HALC on the CHAIR benchmark, demonstrating the effectiveness of our approach. We have incorporated these results into Table 2 and Table C1 of the revised paper.

---

> ### Author Response · Authors · 2024-11-21
> **Response to Reviewer aaN5 (Part 2/3)**
>
> **Comment (3)**: “*No inference costs are listed and compared, which is important for LLM decoding strategies.*”
>
> **Response (3)**: Thanks for your thoughtful comments. We hope we have already addressed this in **Response (1)**. Please let us know if you have any further feedback.
>
> ---
>
> **Comment (4)**: “*Experiments only conducted two LVLM backbones. More results like on QwenVL are better performed.*”
>
> **Response (4)**: Thank you for pointing out this. Following your comments, we have conducted additional experiments to evaluate the effectiveness of our approach using the Qwen-VL model. First, we present the results on the POPE benchmark using the MS-COCO dataset:
>
> ### POPE Benchmark with Qwen-VL Backbone:
> **Random Setup**
> | Method    | Acc. $\uparrow$ | Prec. $\uparrow$ | Rec. $\uparrow$ | F1 $\uparrow$  |
> |-----------|---------|----------|--------|-------|
> | Regular   | 87.43   | 93.56    | 80.40   | 86.48 |
> | VCD       | 88.80    | 93.89    | 83.00   | 88.11 |
> | M3ID      | 89.83   | 95.44    | 83.65  | 89.17 |
> | RITUAL    | 89.47   | **96.32**    | 82.07  | 88.62 |
> | **Ours**  | **89.73** | 93.19  | **85.73** | **89.31** |
>
> **Popular Setup**
> | Method    | Acc. $\uparrow$ | Prec. $\uparrow$ | Rec. $\uparrow$ | F1 $\uparrow$  |
> |-----------|---------|----------|--------|-------|
> | Regular   | 84.70    | 88.24    | 80.07  | 83.96 |
> | VCD       | 85.13   | 87.27    | 82.27  | 84.69 |
> | M3ID      | 86.27   | 89.19    | 82.53  | **85.73** |
> | RITUAL    | 84.57   | 84.09    | **85.27** | 84.67 |
> | **Ours**  | **86.5** | **89.87** | 81.93  | 85.71 |
>
> **Adversarial Setup**
> | Method    | Acc. $\uparrow$ | Prec. $\uparrow$ | Rec. $\uparrow$ | F1 $\uparrow$  |
> |-----------|---------|----------|--------|-------|
> | Regular   | 79.83   | 80.13    | 79.33  | 79.73 |
> | VCD       | 81.33   | 80.60     | 82.53  | 81.55 |
> | M3ID      | 82.03   | 81.47    | **82.93** | 82.19 |
> | RITUAL    | 82.80    | 83.15    | 82.27  | 82.71 |
> | **Ours**  | **83.47** | **84.49** | 81.52  | **82.98** |
>
> We have provided the full results in Table 1 of the revised paper. Please kindly refer to it for the results on A-OKVQA and GQA datasets.
>
> On the MME-Hallucination benchmark, our approach also achieves state-of-the-art results on the Qwen-VL method, as presented below. We have also incorporated this into Table C2 of the Appendix (due to the page limit).
>
>
> ### MME-Hallucination Result with Qwen-VL backbone
>
> | Method   | Existence $\uparrow$      | Count $\uparrow$         | Position $\uparrow$       | Color $\uparrow$         | Total Score $\uparrow$    |
> |----------|---------------------------|--------------------------|---------------------------|--------------------------|---------------------------|
> | Regular  | 155.00 (±3.54)           | 127.67 (±13.36)         | 131.67 (±7.73)           | 173.00 (±9.75)          | 587.33 (±31.06)          |
> | VCD      | 156.00 (±6.52)           | 131.00 (±6.19)          | 128.00 (±3.61)           | 181.67 (±5.14)          | 596.67 (±11.61)          |
> | M3ID     | 178.33 (±2.89)           | 143.33 (±2.89)          | 150.00 (±2.89)           | 175.00 (±5.00)          | 646.66 (±8.50)           |
> | RITUAL   | 178.33 (±2.89)           | 142.22 (±16.19)         | **156.66** (±2.89)           | **178.33** (±2.89)          | 655.55 (±14.99)          |
> | Ours     | **180.00** (±0.00)           | **148.89** (±6.74)          | 155.00 (±7.64)           | **178.33** (±2.89)          | **662.22** (±4.37)           |
>
> In summary, these results demonstrate the general effectiveness of our approach across various LVLMs.

---

> ### Author Response · Authors · 2024-11-21
> **Response to Reviewer aaN5 (Part 3/3)**
>
> **Comment (5)**: “*How about using the middle step of Stable Diffusion to provide generative feedback and reduce the computation costs? How about integrating the generative feedback in the instruction tuning phase? It will avoid inference costs.*”
>
> **Response (5)**: Thank you for your insightful comments. Based on your feedback, we have implemented two strategies to accelerate our approach, including utilizing the middle step of Stable Diffusion, which has proven effective. We sincerely appreciate your constructive input!
>
> Specifically, we propose two strategies to improve efficiency: (1) reducing the number of inference steps in the diffusion process and (2) limiting the length of the initial response.
>
> - **Reducing Diffusion Inference Steps**. To improve the response generation speed, we can reduce the number of diffusion steps. In the following table, we report the performance on the CHAIR benchmark after reducing the diffusion inference steps in the model. By reducing the diffusion inference steps from 50 to 10, the average latency decreases by 2.85 seconds per instance, while the performance on CHAIR remains robust.
>
>     | Diffusion Steps | Avg. Latency $\downarrow$ | CHAIR$_S$ $\downarrow$ | CHAIR$_I$     $\downarrow$ |
>     | --------------- | ------------------------- | -------------------- | -------------------- |
>     | 50   | 13.89 s   | 48.8    | 14.6|
>     | 30   | 12.56 s   | 48.9   | 14.7   |
>     | 20 | 11.87 s   | 49.2   | 14.8    |
>     | 10    | 11.04 s    | 48.8   | 14.9    |
>
> - **Restricting Length of Initial Response**. Our method involves two queries to the LVLM for self-correcting decoding. To enhance efficiency, we can limit the length of the initial response. In the following table, we present the efficiency and CHAIR performance results after decreasing the maximum token limit for the initial response. We can see that reducing the maximum number of tokens in the initial response from 128 to 96 decreases the latency by 0.72 seconds per instance while maintaining competitive performance. However, further reductions result in performance degradation, as a shorter initial response fails to adequately cover the entire scene, limiting its ability to generate an image that effectively reflects and mitigates hallucinations.
>
>     | # Tokens | Avg. Latency $\downarrow$ | CHAIR$_S$ $\downarrow$ | CHAIR$_I$ $\downarrow$ |
>     | -------- | ------------------------- | -------------------- | -------------------- |
>     | 128 | 13.89 s  | 48.8  | 14.6   |
>     | 96  | 13.17 s  | 48.8 | 14.9  |
>     | 64   | 12.20 s | 49.5 | 14.8 |
>     | 32  | 11.33 s | 51.2  | 14.9  |
>
>     Note that these two strategies are not conflicting; instead, they are complementary. Setting the diffusion steps to 10 and limiting the maximum number of tokens in the initial response to 96 further reduces the inference latency to 10.21 seconds per instance while maintaining robust performance.
>
> Besides, incorporating generative feedback into the instruction tuning phase is also an interesting idea. We have added a discussion about this in Appendix E of the revised paper. We will examine this idea in greater detail and explore it further in our future work.
>
> ---
>
>
> **Comment (6)**: “*Why not conduct GPT4/GPT4-V analysis like VCD [r1]?*”
>
> **Response (6)**: Thank you for pointing out this. In the manuscript, we have included GPT-4V analysis in some of our qualitative examples. Following your comments, we conduct an additional GPT-4V evaluation on the open-ended captioning tasks in the LLaVA-Bench following the VCD evaluation protocol. The settings and experimental details can be found in Appendix D.1. Note that in this setting, GPT-4V only evaluates two responses at a time. Therefore, we divide the experiments into three parts: Ours vs Regular, Ours vs VCD, and Ours vs M3ID. We report the results as follows.
>
> ### GPT-4V Evaluation with LLaVA-1.5 backbone:
>
> | Model   | Accuracy | Detailedness |
> | ------- | -------- | ------------ |
> | Regular | 2.88     | 3.29         |
> | Ours    | **4.29** | **4.54**     |
> | VCD     | 3.62     | 3.83         |
> | Ours    | **4.04** | **4.38**     |
> | M3ID    | 3.88     | 4.08         |
> | Ours    | **4.04** | **4.29**     |
>
> ### GPT-4V Evaluation with InstructBLIP backbone:
>
> | Model   | Accuracy | Detailedness |
> | ------- | -------- | ------------ |
> | Regular | 3.42     | 3.96         |
> | Ours    | **4.38** | **4.79**     |
> | VCD     | 3.71     | 4.21         |
> | Ours    | **4.17** | **4.58**     |
> | M3ID    | 4.00     | 4.33         |
> | Ours    | **4.08** | **4.50**     |
>
> From the results, we can observe that our method achieves higher accuracy and detailedness in generated responses compared to regular decoding, VCD, and M3ID across both evaluated LVLMs. We have incorporated these results into Table 4 of the revised paper.
>
> ---
>
>
> If you have additional comments or concerns, please let us know and we will be more than happy to answer.
>
> Best,
>
> Authors

---

> > ### Comment · Reviewer_aaN5 · 2024-11-22
> > **Official Comment by Reviewer aaN5**
> >
> > Thank you for the detailed response! I check the revised paper, it seems that comparisons with Woodpecker and HALC are only conducted in the CHAIR metric.

---

> > > ### Author Response · Authors · 2024-11-23
> > > **Follow-up Response to Reviewer aaN5**
> > >
> > > Dear Reviewer aaN5,
> > >
> > > Thank you for your follow-up comments.
> > >
> > > ---
> > >
> > > Following your comments, we have further evaluated Woodpecker and HALC on the MME and MMBench benchmarks and report the results as follows.
> > >
> > > | **Method** | Existence ↑        | Count ↑            | Position ↑         | Color ↑            | **MME Score** ↑    | **MMBench** ↑ |
> > > | ---------- | ------------------ | ------------------ | ------------------ | ------------------ | ------------------ | ------------- |
> > > | Regular    | 173.75 (±4.79)     | 121.67 (±12.47)    | 117.92 (±3.69)     | 149.17 (±7.51)     | 562.50 (±3.96)     | 64.1          |
> > > | DoLa       | 176.67 (±2.89)     | 113.33 (±10.41)    | 90.55 (±8.22)      | 141.67 (±7.64)     | 522.22 (±16.78)    | 63.8          |
> > > | OPERA      | 183.33 (±6.45)     | 137.22 (±6.31)     | 122.78 (±2.55)     | 155.00 (±5.00)     | 598.33 (±10.41)    | 64.4          |
> > > | VCD        | 186.67 (±5.77)     | 125.56 (±3.47)     | 128.89 (±6.73)     | 139.45 (±12.51)    | 580.56 (±15.13)    | _64.6_        |
> > > | M3ID       | 186.67 (±5.77)     | 128.33 (±10.41)    | _131.67_ (±5.00)   | 151.67 (±20.88)    | 598.11 (±20.35)    | 64.4          |
> > > | RITUAL     | _187.50_ (±2.89)   | _139.58_ (±7.64)   | 125.00 (±10.27)    | _164.17_ (±6.87)   | _616.25_ (±20.38)  | 63.8          |
> > > | Woodpecker | _187.50_ (±2.89)   | 125.00 (±0.00)     | 126.66 (±2.89)     | 149.17 (±17.34)    | 588.33 (±10.00)    | 64.0          |
> > > | HALC       | 183.33 (±0.00)     | 133.33 (±5.77)     | 107.92 (±3.69)     | 155.00 (±5.00)     | 579.58 (±9.07)     | 64.2          |
> > > | Ours       | **188.33** (±2.89) | **150.00** (±7.64) | **133.89** (±3.85) | **172.22** (±3.47) | **644.44** (±9.18) | **65.5**      |
> > >
> > > As we can see from the results, our approach also outperforms Woodpecker and HALC across these two benchmarks, validating the effectiveness of our approach. We have incorporated these results into Table 3 of the revised paper.
> > >
> > > We use the official HALC codebase (https://github.com/BillChan226/HALC) to reproduce and evaluate two models, HALC and Woodpecker, on the CHAIR, MME, and MMBench benchmarks. While we made every effort to evaluate these methods on the POPE benchmark, the HALC paper evaluates the models on the offline POPE benchmark (as detailed in Section 6.1 of [1]), and the code for the standard POPE benchmark evaluation is not provided. This has made reproduction and evaluation particularly challenging within the short rebuttal phase. Additionally, while the paper provides POPE results, we cannot directly include them in our comparisons because our evaluations are based on LLaVA-1.5 and InstructBLIP, whereas their reported results are based on LLaVA and MiniGPT-4. We are actively working on implementing and evaluating these models on the POPE benchmark and will include the results in the final version.
> > >
> > > However, we believe that the comprehensive performance comparisons on the CHAIR, MME, and MMBench benchmarks consistently demonstrate the superiority of the proposed method over HALC and Woodpecker.
> > >
> > > [1] Chen, Z., Zhao, Z., Luo, H., Yao, H., Li, B., & Zhou, J. HALC: Object Hallucination Reduction via Adaptive Focal-Contrast Decoding. In *Forty-first International Conference on Machine Learning*.
> > >
> > > ---
> > >
> > > We hope that our responses have addressed your concerns. Please kindly let us know if you have any further questions.
> > >
> > > Best,
> > >
> > > Authors

---

> > > > ### Comment · Reviewer_aaN5 · 2024-11-23
> > > > **Official Comment by Reviewer aaN5**
> > > >
> > > > Thank you for the detailed response! I decide to raise my score.

---

> > > > > ### Author Response · Authors · 2024-11-23
> > > > > **Thanks for your positive recommendation**
> > > > >
> > > > > Dear Reviewer aaN5,
> > > > >
> > > > > We sincerely appreciate the time and effort you invested in reviewing our manuscript. We greatly appreciate your positive recommendation!
> > > > >
> > > > > Best,
> > > > >
> > > > > Authors

---

### Official Review · Reviewer_XJFu · 2024-10-29

**Soundness:** 3
**Presentation:** 3
**Contribution:** 3
**Rating:** 6
**Confidence:** 4

**Summary:**

This paper proposes to mitigate hallucinations in LVLMs by introducing valuable feedback from text-to-image generative models. Then a training-free decoding algorithm is introduced to integrate information from the original images and feedback from text-to-image generative models. Experiments on five benchmarks and two LVLMs demonstrate the effectiveness of the proposed method.

**Strengths:**

1.	The idea of leveraging text-to-image generative models for LVLM hallucination mitigation is novel and interesting.
2.	The paper is well-written and easy to follow.
3.	Extensive experiments on multiple benchmarks demonstrate the effectiveness of the proposed method.

**Weaknesses:**

1.	The effectiveness of the proposed method is heavily influenced by the quality and realism of the generated images. So the authors should perform more experiments and analysis on the quality of the generated images (both quantitative and qualitative, especially for long captions and real images).
2.	Since the model relies on an addition diffusion model for inference, additional inference overhead should be discussed.
3.	Experiments are limited to two LVLMs, experiments on more recent LVLMs (e.g., QWenVL, MiniCPMV, etc.) should be conducted.
4.	It’s better to perform experiments on multiple object hallucination [1].
5.	The approach involves various hyperparameters that require careful tuning to achieve optimal performance, which could make practical implementation challenging.

[1] Chen X, Ma Z, Zhang X, et al. Multi-object hallucination in vision-language models[J]. arXiv preprint arXiv:2407.06192, 2024.

**Questions:**

Please see Weakness.

---

> ### Author Response · Authors · 2024-11-21
> **Response to Reviewer XJFu (Part 1/3)**
>
> Dear Reviewer XJFu,
>
> We really appreciate your thorough review of our paper. We address the raised concerns and questions below.
>
> ---
>
> **Comment (1)**: “*The effectiveness of the proposed method is heavily influenced by the quality and realism of the generated images. So the authors should perform more experiments and analysis on the quality of the generated images (both quantitative and qualitative, especially for long captions and real images)*.”
>
> **Response (1)**: Thank you for your thoughtful feedback. Following your comments, we have evaluated the quality of the generated images on all 4 subsets on the MME benchmark using CLIPScore [1]. Specifically, we utilize the CLIP backbone with ViT-B/32 backbone for our evaluation. We list the results below:
>
> | MME Subset     | Existence | Count | Position | Color |
> | -------------- | --------- | ----- | -------- | ----- |
> | Avg. CLIPScore | 31.34     | 30.69 | 30.09    | 31.69 |
>
> As we can see from the table, our text-to-image generative model (specifically, SD-v1.5) achieves an average CLIPScore of over 30 across all subsets. For comparison, the advanced DALL-E 3 model achieves a score of 32.0, while DALL-E 2 achieves 31.4 [2]. These results highlight the capability of our model to generate high-quality images that closely align with the initial response. We have incorporated this discussion into Appendix C.9.
>
> Additionally, in Figure D3-D6, we provide qualitative evaluations of the images generated by the generative model, including both success and failure cases, across all four subsets of the MME benchmark.  These evaluations provide deeper insights into the effectiveness of the generative models. Our results show that, despite occasional failure cases, the generative model consistently produces high-quality and realistic images that accurately visualize the initial response, providing effective self-feedback.
>
> [1] Hessel, J., Holtzman, A., Forbes, M., Le Bras, R., & Choi, Y. (2021, November). CLIPScore: A Reference-free Evaluation Metric for Image Captioning. In *Proceedings of the 2021 Conference on Empirical Methods in Natural Language Processing* (pp. 7514-7528).
>
> [2] Betker, J., Goh, G., Jing, L., Brooks, T., Wang, J., Li, L., ... & Ramesh, A. (2023). Improving image generation with better captions. https://cdn.openai.com/papers/dall-e-3.pdf

---

> ### Author Response · Authors · 2024-11-21
> **Response to Reviewer XJFu (Part 2/3)**
>
> **Comment (2)**: “*Since the model relies on an addition diffusion model for inference, additional inference overhead should be discussed*.”
>
> **Response (2)**: Thanks for pointing out this. We will respond from two aspects:
>
> - **Efficiency Comparison**. Following your comments, we compare the efficiency of our approach with other state-of-the-art approaches such as OPERA, Woodpecker, and HALC. Specifically, we conduct this comparison on the CHAIR benchmark, with the maximum token length set to 128. The results are as follows:
>
>     | Method     | Avg. Latency $\downarrow$ | GPU Memory $\downarrow$  | CHAIR$_S$ $\downarrow$ |
>     | ---------- | ------------------------- | ------------------------ | ---------------------- |
>     | Regular    | 3.44 s ($\times$1.00)     | 15778 MB ($\times$1.00)  | 55.0  |
>     | VCD        | 6.91 s ($\times$2.01)     | 16634 MB ($\times$1.05) | 54.4 |
>     | OPERA      | 24.70 s ($\times$7.18)    | 22706 MB ($\times$1.44)  | 52.6  |
>     | Woodpecker | 10.68 s ($\times$3.10)    | 22199 MB ($\times$1.41)  | 57.6 |
>     | HALC       | 22.61 s ($\times$6.51)    | 23084 MB ($\times$1.46)  | 51.0 |
>     | Ours       | 13.89 s ($\times$4.04)    | 19119 MB  ($\times$1.21) | 48.8  |
>
>     Our approach involves two queries and incorporates a text-to-image generation model to mitigate hallucinations, resulting in a 4.04$\times$ increase in latency and a 1.21$\times$ increase in GPU memory usage. Compared to other approaches, while our method is slower than regular decoding and contrastive decoding-based methods, it demonstrates efficiency advantages over OPERA and HALC. Note that our approach also outperforms the compared methods in performance on the CHAIR benchmark.
>
> - **Strategies to Speed Up Our Approach**. To further enhance the efficiency of our approach, we propose two strategies to accelerate our approach: limiting the length of the initial response and reducing the number of inference steps in the diffusion process.
>
>     **(1) Reducing Diffusion Inference Steps**. In the following table, we report the performance on the CHAIR benchmark after reducing the diffusion inference steps in the model. By reducing the diffusion inference steps from 50 to 10, the average latency decreases by 2.85 seconds per instance, while the performance on CHAIR remains robust.
>
>     | Diffusion Steps | Avg. Latency $\downarrow$ | CHAIR$_S$ $\downarrow$ | CHAIR$_I$     $\downarrow$ |
>     | --------------- | ------------------------- | -------------------- | -------------------- |
>     | 50   | 13.89 s   | 48.8    | 14.6|
>     | 30   | 12.56 s   | 48.9   | 14.7   |
>     | 20 | 11.87 s   | 49.2   | 14.8    |
>     | 10    | 11.04 s    | 48.8   | 14.9    |
>
>     **(2) Restricting Length of Initial Response**. In the following table, we present the efficiency and CHAIR performance results after decreasing the maximum token limit for the initial response. We can see that reducing the maximum number of tokens in the initial response from 128 to 96 decreases the latency by 0.72 seconds per instance while maintaining competitive performance. However, further reductions result in performance degradation, as a shorter initial response fails to adequately cover the entire scene, limiting its ability to generate an image that effectively reflects and mitigates hallucinations.
>
>     | # Tokens | Avg. Latency $\downarrow$ | CHAIR$_S$ $\downarrow$ | CHAIR$_I$ $\downarrow$ |
>     | -------- | ------------------------- | -------------------- | -------------------- |
>     | 128 | 13.89 s  | 48.8  | 14.6   |
>     | 96  | 13.17 s  | 48.8 | 14.9  |
>     | 64   | 12.20 s | 49.5 | 14.8 |
>     | 32  | 11.33 s | 51.2  | 14.9  |
>
>     Note that these two strategies are not conflicting; instead, they are complementary. Setting the diffusion steps to 10 and limiting the maximum number of tokens in the initial response to 96 further reduces the inference latency to 10.21 seconds per instance while maintaining robust performance.
>
> We have also added these discussions and additional experiments to Section 4.4 and Appendix C.8 of the revised paper. Again, thank you for your valuable feedback to improve our paper.

---

> ### Author Response · Authors · 2024-11-21
> **Response to Reviewer XJFu (Part 3/3)**
>
> **Comment (3)**: “*Experiments are limited to two LVLMs, experiments on more recent LVLMs (e.g., QWenVL, MiniCPMV, etc.) should be conducted*.”
>
> **Response (3)**: Thank you for pointing out this. Following your comments, we have conducted additional experiments to evaluate the effectiveness of our approach using the Qwen-VL model. First, we present the results on the POPE benchmark using the MS-COCO dataset:
>
> **Random Setup**
> | Method | Acc. $\uparrow$ | Prec. $\uparrow$ | Rec. $\uparrow$ | F1 $\uparrow$  |
> |-----------|---------|----------|--------|-------|
> | Regular| 87.43 | 93.56| 80.40   | 86.48 |
> | VCD | 88.80| 93.89 | 83.00   | 88.11 |
> | M3ID | 89.83 | 95.44    | 83.65  | 89.17 |
> | RITUAL | 89.47   | **96.32**    | 82.07  | 88.62 |
> | **Ours**  | **89.73** | 93.19  | **85.73** | **89.31** |
>
> **Popular Setup**
> | Method    | Acc. $\uparrow$ | Prec. $\uparrow$ | Rec. $\uparrow$ | F1 $\uparrow$  |
> |-----------|---------|----------|--------|-------|
> | Regular | 84.70    | 88.24    | 80.07  | 83.96 |
> | VCD | 85.13   | 87.27    | 82.27  | 84.69 |
> | M3ID | 86.27   | 89.19    | 82.53  | **85.73** |
> | RITUAL | 84.57   | 84.09    | **85.27** | 84.67 |
> | **Ours**  | **86.5** | **89.87** | 81.93  | 85.71 |
>
> **Adversarial Setup**
> | Method    | Acc. $\uparrow$ | Prec. $\uparrow$ | Rec. $\uparrow$ | F1 $\uparrow$  |
> |-----------|---------|----------|--------|-------|
> | Regular   | 79.83 | 80.13  | 79.33  | 79.73 |
> | VCD  | 81.33 | 80.60  | 82.53  | 81.55 |
> | M3ID | 82.03 | 81.47 | **82.93** | 82.19 |
> | RITUAL | 82.80 | 83.15 | 82.27  | 82.71 |
> | **Ours**  | **83.47** | **84.49** | 81.52  | **82.98** |
>
> We have provided the full results in Table 1 of the revised paper. Please kindly refer to it for the results on A-OKVQA and GQA datasets.
>
> On the MME-Hallucination benchmark, our approach also achieves state-of-the-art results on the Qwen-VL method, as presented below. We have also incorporated this into Table C2 of the Appendix (due to the page limit).
>
>
> | Method   | Existence $\uparrow$    | Count $\uparrow$| Position $\uparrow$ | Color $\uparrow$ | Total Score $\uparrow$    |
> |----------|---------------------------|--------------------------|---------------------------|--------------------------|---------------------------|
> | Regular  | 155.00 (±3.54)  | 127.67 (±13.36)  | 131.67 (±7.73)   | 173.00 (±9.75)  | 587.33 (±31.06)  |
> | VCD | 156.00 (±6.52) | 131.00 (±6.19)  | 128.00 (±3.61) | 181.67 (±5.14)  | 596.67 (±11.61)          |
> | M3ID  | 178.33 (±2.89) | 143.33 (±2.89)  | 150.00 (±2.89)    | 175.00 (±5.00)    | 646.66 (±8.50)           |
> | RITUAL   | 178.33 (±2.89)   | 142.22 (±16.19) | **156.66** (±2.89)| **178.33** (±2.89)  | 655.55 (±14.99)    |
> | Ours     | **180.00** (±0.00)  | **148.89** (±6.74)  | 155.00 (±7.64) | **178.33** (±2.89)   | **662.22** (±4.37)  |
>
>
> In summary, these results demonstrate the general effectiveness of our approach across various LVLMs.
>
>
> ---
>
> **Comment (4)**: “*It’s better to perform experiments on multiple object hallucination [1]*.”
>
> **Response (4)**: Thank you for your constructive suggestion and for bringing this interesting benchmark to our attention. We address your comment from two perspectives:
> - We would like to highlight that our extensively evaluated benchmarks already include scenarios involving multi-object hallucinations, such as in open-ended generation tasks like CHAIR and the GPT-4V evaluation on LLaVA-Bench.
> - Given that this benchmark is very new, it has been challenging for us to adapt our codebase and conduct evaluations on it within the short rebuttal phase. However, we have acknowledged this benchmark and included it in our discussion on future work, as detailed in Appendix E of the revised paper. We greatly appreciate your valuable feedback and look forward to exploring this benchmark in future research.
>
> ---
>
>
> **Comment (5)**: “*The approach involves various hyperparameters that require careful tuning to achieve optimal performance, which could make practical implementation challenging*.”
>
> **Response (5)**: Thanks for your thoughtful feedback. While our approach introduces several hyperparameters to effectively leverage generative feedback, the performance of our approach remains robust across a reasonable range of hyperparameter settings. As shown in our ablation studies (Section 4.3 and Appendix C), on the POPE benchmark using the COCO random subset, setting hyperparameters within ranges of $\alpha_1 \in [0, 4]$, $\alpha_2 \in [1, 3]$, $\gamma \in [0, 1]$, and $\beta \in [0.05, 0.5]$ achieves an accuracy of >87.5% and consistently outperforms VCD and M3ID. Please also be assured that we will make the code implementation publicly available and provide detailed instructions upon acceptance to facilitate easier reproduction.
>
> ---
>
>
> We hope that our responses have addressed your concerns. If you have additional comments or concerns, please let us know and we will be more than happy to answer.
>
> Best,
>
> Authors

---

> > ### Comment · Reviewer_XJFu · 2024-11-23
> >
> > Thanks for your responses, I have raised my score.

---

> > > ### Author Response · Authors · 2024-11-23
> > > **Thanks for your positive recommendation**
> > >
> > > Dear Reviewer XJFu,
> > >
> > > Thank you for taking the time to review our paper. We sincerely appreciate your thoughtful feedback and positive recommendation!
> > >
> > > Best,
> > >
> > > Authors

---

### Author Response · Authors · 2024-11-21
**General Response**

Dear AC and Reviewers,

We are sincerely grateful to you all for dedicating time and efforts in providing these detailed and thoughtful reviews, which helped us to improve the quality of our paper. We have also carefully revised the paper based on your thoughtful feedback. For your convenience, we have highlighted all the revisions made compared to the initial version in blue.

Here, apart from the point-by-point responses to each reviewer, we would like to summarize the contributions of this work and highlight our new results added during the rebuttal phase.

---

We are delighted that the reviewers appreciate and recognize the following strengths and contributions of this work:
- The idea of leveraging text-to-image generative models for LVLM hallucination mitigation is novel and interesting. Providing generative feedback to mitigate hallucinations is straightforward and reasonable. **[XJFu, aaN5]**
- The paper is well-motivated: The study finds that text-to-image generative models effectively identify hallucinations, which gives a strong empirical foundation of the proposed decoding approach.  **[QadT, prZa]**
- Extensive experiments on multiple benchmarks demonstrate the effectiveness of the proposed method. **[All Reviewers]**
- The paper is well-written and easy to follow. **[XJFu, QadT, aaN5]**

---

In this rebuttal, we have included the following discussions and experiments to address reviewers’ comments:
- We compare the efficiency of our approach with state-of-the-art methods in Table 7 and discuss two acceleration strategies in Section C.8 of the revised paper.
- Additional experiments comparing our approach to other methods on the Qwen-VL model for the POPE and MME-Hallucination benchmarks are presented in Table 1 and Table C2.
- Quantitative and qualitative evaluations of generated image quality across all four subsets of the MME benchmark are detailed in Appendix C.9 and Figures D3–D6.
- Additional performance comparisons on the MMBench benchmark are provided in Table 3 and Table C3.
- We include a GPT-4V evaluation on open-ended captioning tasks in LLaVA-Bench, following the VCD evaluation protocol, with results shown in Table 4.
- Additional performance comparisons on the CHAIR benchmark, with the maximum token limit set to 128, as well as comparisons with HALC and Woodpecker, are presented in Table 2 and Table C1.
- Additional results for the POPE benchmark under greedy decoding are reported in Table C4.

---

Again, thank you for your time in reviewing our work! If you have additional comments or concerns, please let us know and we will be more than happy to answer.

Best,

Authors

---

### Author Response · Authors · 2024-12-02
**Thank you for your constructive feedback. Please feel free to share any last-minute questions.**

Dear Reviewers,

We sincerely appreciate your efforts in providing thoughtful reviews and actively participating in the discussions. As the discussion period concludes today, December 2nd, at midnight AoE, please let us know if you have any last-minute questions or concerns. We would be delighted to provide any additional details or clarifications.

Best,

Authors

---

### Meta-Review · Area_Chair_FCZw · 2024-12-17

**Metareview:**

The paper proposes a method called self-correcting decoding with generative feedback (DeGF), which explores self-feedback for correcting hallucinations of LVLMs at both the response and token levels. The proposed method is well-motivated and technically sound. The experimental results are extensive and convincing. The rebuttal successfully addressed the major concerns of effectiveness (additional inference overhead) and additional baselines raised by the reviewers. At the end of the rebuttal, all reviewers agreed to accept this paper, putting the paper above the bar.

**Additional Comments On Reviewer Discussion:**

The rebuttal successfully addressed the major concerns of effectiveness (additional inference overhead) and additional baselines raised by the reviewers. At the end of the rebuttal, all reviewers agreed to accept this paper, putting the paper above the bar.

---

### Decision · Program_Chairs · 2025-01-22

Accept (Poster)